

# Effects of mesoscale eddies on behavior of an oil spill resulting from an accidental deepwater blowout in the Black Sea: an assessment of the environmental impacts

Konstantin A. Korotenko

Physical Oceanography, Shirshov Institute of Oceanology, RAS, Moscow, Russian Federation

## ABSTRACT

Because of the environmental sensitivity of the Black Sea, as a semi-enclosed sea, any subsea oil spill can cause destructive impacts on the marine environment and beaches. Employing numerical modeling as a prediction tool is one of the most efficient methods to understand oil spill behavior under various environmental forces. In this regard, a coupled circulation/deepsea oil spill model has been applied to the Black Sea to address the behavior of the oil plume resulting from a representative hypothetical deepwater blowout. With climatological forcing, the hydrodynamic module based on DieCAST ocean circulation model realistically reproduces seasonally-varying circulation from basin-scale dominant structures to meso- and sub-mesoscale elements. The oil spill model utilizes pre-calculated DieCAST thermo-hydrodynamic fields and uses a Lagrangian tracking algorithm for predicting the displacement of a large number of seeded oil droplets, the sum of which forms the rising oil plume resulting from a deepwater blowout. Basic processes affecting the transport, dispersal of oil and its fate in the water column are included in the coupled model. A hypothetical oil source was set at the bottom, at the northwestern edge of the Shatsky Ridge in the area east of the Crimea Peninsula where the oil exploration/development is likely to be planned. Goals of the study are to elucidate the behavior of the subsea oil plume and assess scales of contamination of marine environment and coastlines resulting from potential blowouts. The two 20-day scenarios with the oil released by a hypothetical blowout were examined to reveal combined effects of the basin-scale current, near-shore eddies, and winds on the behavior of the rising oil plume and its spreading on the surface. Special attention is paid to the Caucasian near-shore anticyclonic eddy which is able to trap surfacing oil, detain it and deliver it to shores. The length of contaminated coastlines of vulnerable Crimean and Caucasian coasts are assessed along with amounts of oil beached and deposited.

# INTRODUCTION

Offshore exploration and development of oil resources as well as the industrial exploitation of oil reserves pose a great threat to the marine environment and venerable beaches.

Corresponding author
Konstantin A. Korotenko,
kkorotenko@gmail.com

For the semi-enclosed Black Sea, such activity can result in lasting damage to the environment and fragile habitat.

These studies were undertaken to improve understanding of how the mesoscale circulation of the Black Sea along with its basin-scale Rim Current (RC) might effect on processes of the transport and dispersal of oil spilled by a deepwater oil well as the result of its accidental damage. Employing numerical simulation is one of the easiest and most effective prediction tools for understanding the oil spill behavior under various environmental forces. The knowledge of the scale of a possible disaster allows the coastguard efforts better focusing to identify preliminary steps toward such an event, decreasing the lead time available for response and mitigation efforts.

Potentially, exploration and development of oil are associated with contamination of the environment as a result of possible accidental spills. Such a problem became too apparent in spring 2010, when about 578,000 tons (3,635,000 barrels) of oil was released into the Gulf of Mexico (GoM) during the almost 3-month catastrophic deepwater blowout following the tragic Deepwater Horizon (DWH) oil rig explosion on April 20, 2010 (*Liu et al., 2011*; *Socolofsky, Adams & Sherwood, 2011*; *North et al., 2011*; *Lavrova & Kostianoy, 2011*; *Paris et al., 2012*; *Le Hénaff et al., 2012*; *Korotenko et al., 2013*; *Dietrich et al., 2014*; *Fingas, 2017*). According to the satellite images, the spill has directly impacted 180,000 km$^2$ of the GoM. However, the anticipated disastrous downstream effects did not materialize and, fortunately, no oil related to the DWH source was not reported along the South Florida coastal areas or in the Atlantic Ocean (www.nytimes.com/interactive/2010/05/01/us/20100501-oil-spill-tracker.html).

The DWH accident presented the challenge to oil spill modelers to realistically predict the behavior of oil spilled by a deepwater blowout (*Camilli et al., 2010*; *Paris et al., 2012*). Many efforts were undertaken to improve numerical models describing the structure of an oil plume rising from a deepwater oil well, the formation and movement of an oil slick on the ocean surface as result of surfacing oil (*Liu et al., 2011*; *Socolofsky, Adams & Sherwood, 2011*; *North et al., 2011*; *Korotenko et al., 2013*). A well-validated ocean circulation model coupled with an oil spill model, in addition to the implementation of specific algorithms describing the fate of oil, should realistically predict the behavior and transformation of the oil plume (*Mariano et al., 2011*; *Korotenko et al., 2013*; *North et al., 2015*). In modeling oil slicks, an important issue is to drive the computation model with realistic winds since misrepresenting the details of the local wind forcing leads to errors in predicting the oil slick behavior and the subsequent distribution of oil concentration (*Caratelli, Dentale & Reale, 2011*; *Le Hénaff et al., 2012*; *Korotenko et al., 2013*).

The catastrophe in the GoM has revealed a very serious threat posed to the marine environment by exploration and development of deepwater oil resources, and despite best efforts to prevent such disastrous events, they seem to be inevitable due to a number of reasons, whether man-made or natural. In this connection, the growing activity in different parts of the oceanic shelf raises a serious environmental concern regarding possible consequences for water bodies where drilling is going on or planned. The semi-enclosed Black Sea suffers from strong ecological disequilibria caused by pollution arising from many contaminants, atmospheric deposition and occasional accidents at the sea.

Among the major contaminants, oil residues have a particular concern. In the past, accidental oil spills resulting from collisions and groundings of oil tankers as well as from accidental spills at oil terminals and damages to coastal pipelines have the major potential environmental impact (*Stoyanov, Dorogan & Jelescu, 1999*; *Korotenko, Bowman & Dietrich, 2003*). In the last decade, intense explorations and developments of deepwater oil resources discovered on the continental shelf of the Black Sea pose a greater threat and risks for the marine environment and coastline than was previously recognized (*Robinson et al., 1996*; *Ergün, Dondurur & Cifci, 2002*; *Egorov et al., 2003*; *Akhmetzhanov et al., 2007*; *Körber et al., 2014*).

According to recent observations, many oil/gas seepage areas were discovered along the Bulgarian continental shelf and more than 6,000 individual seeps are identified offshore Bulgaria. Some 10,000 of seepage are reported to exist within the Georgian shelf (*Körber et al., 2014*). Several areas of active gas venting and oil seeps are also discovered in Romanian, Ukrainian, and Turkish waters. More than 500 gas plumes are documented by echo-sounding along the shelf break of the Western and North-Western part of the Black Sea. Abundant gas seepage have been found around the edge of the basin in water depth down to 800 m along the shelf break and active faults in the shelf areas, especially along the frontal lines of Balkanides, Crimea, and Great Caucasus, in the northwestern shelf where several oil and gas fields in the Bulgarian, Ukrainian, and Romanian shelves are exploiting (https://www.offshoreenergytoday.com/tag/black-sea/). To confirm the Black Sea's reserves, an ultra-deepwater well was recently drilled off the coast of Turkey (*Ergün, Dondurur & Cifci, 2002*). It was performed under the framework of exploration plans and Turkish experts estimated that the Black Sea's recoverable reserves of crude oil contain about 10 billion barrels. Oil and gas reserves in Georgian, Russian, and Ukrainian offshore sectors are also planned to develop in near future to boost the oil production in this region. In this connection, an important issue for environmentalists now is to control all activities related to offshore oil and gas exploration in the Black Sea and assessing the risks posed by these activities to the marine ecosystem. It is worthwhile to mention the EU EMODNET Black Sea Checkpoint monitoring system (http://emodnet-blacksea.eu/) in the framework of which the sub-system "Oil Platform Leaks" aims to monitor oil spills over the Black Sea. This sub-system provides oil spill trajectory monitoring and assessments of environmental and coastal impacts. Another monitoring system that also should be mentioned and being of relevance for the study presented below, is the "MyOcean" (*Zodiatis et al., 2012*) providing identification of the exact location of the spill, predicting of the direction of the slick drift and its final location as well as arrival time, etc. Such predictions can greatly assist the agencies, related to marine safety, for reducing the impact on the marine environment that may arise from pollution incidents.

In the presented work, the attention was focused on the northeastern part of the Black Sea because together with intensive oil drillings on shelves of Turkey, Romania, Ukraine, and Georgia that have already started, there are wider plans to start exploration offshore deepwater oil drilling in the Russian sector of the Black Sea in the region of Shatsky Ridge where offshore oil-rich deposits were also discovered (*Laverov, 2003*).

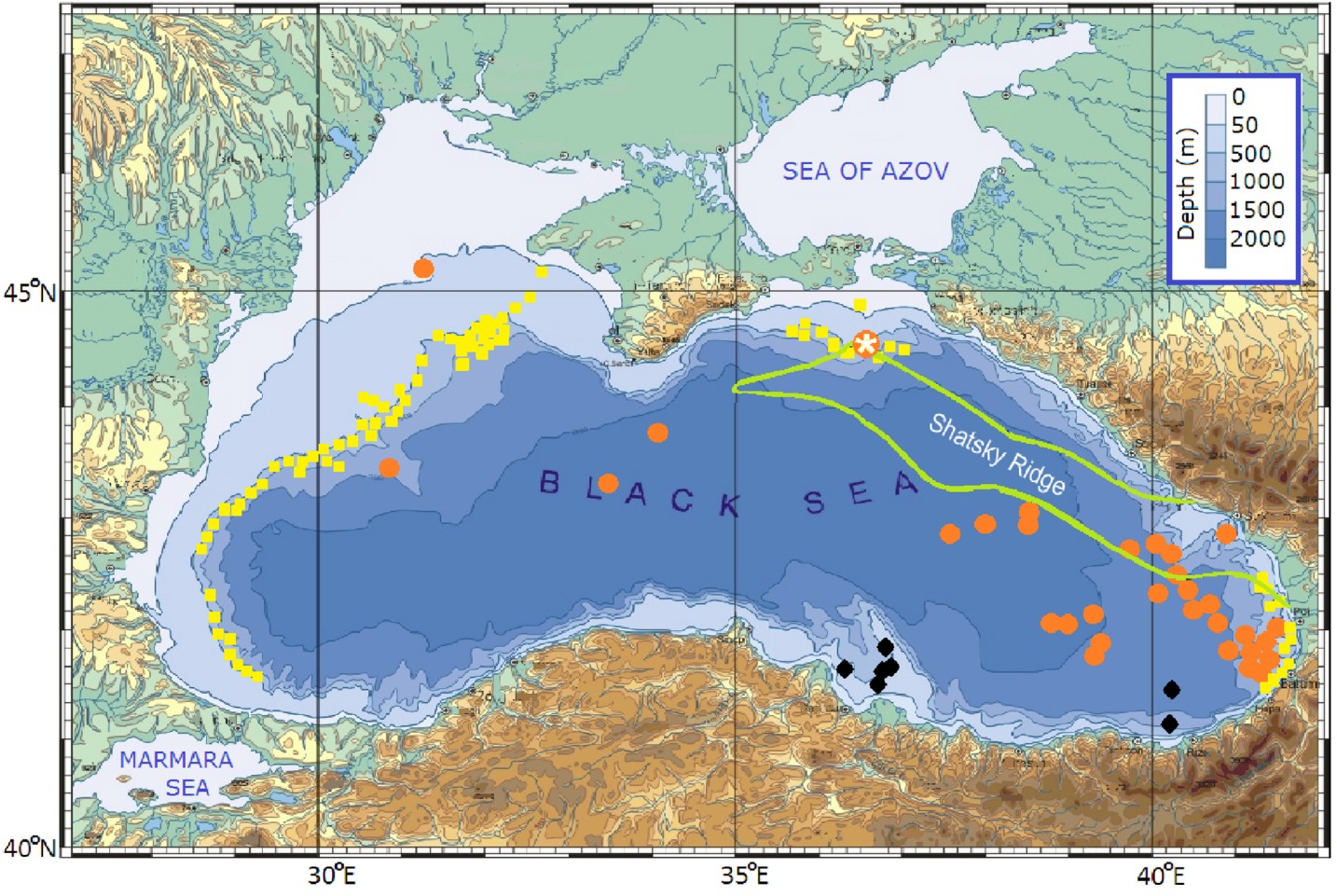

**Figure 1 The Black Sea bottom topography and composite map of areas, where the presence gas and oil was revealed (modified from *Korotenko (2016)*).** Red dots indicate origins of oil slicks (*Robinson et al., 1996*), yellow squares indicate distributions of gas flares (*Egorov et al., 2003*) and black diamonds indicate gas seepages (*Ergün, Dondurur & Cifci, 2002*). The white asterisk inside of the red dot denotes the location of hypothetical deepwater oil blowout over the northwestern edge of the Shatsky Ridge delineated by the green line.

Figure 1 shows the Black Sea bottom topography and composite (obtained in different research cruises) map of discovered offshore gas and oil resources (*Robinson et al., 1996*; *Ergün, Dondurur & Cifci, 2002*; *Egorov et al., 2003*; *Akhmetzhanov et al., 2007*; *Körber et al., 2014*). The structure of the cyclonic basin-scale circulation along the Caucasian and Crimean coasts is well investigated and even rough estimations reveal that the serious environmental risks may arise as the result of possible accidents during offshore oil exploration and development in this region of Black Sea.

Major anthropogenic incidents, when oil was spilled at the sea surface or brought there from deepsea, often resulted in the formation of massive oil slicks, extending for hundreds of kilometers. In such cases, it is impossible to save the entire coastline. Therefore, the protection plans should focus on the most important and vulnerable shorelines.

Being effective prediction tools, numerical simulations widely used to study the behavior of oil spills of different origin. It should be emphasized that despite many efforts
were undertaken for prediction transport and dispersal of oil pollution in the Black Sea, by now, none of the oil spill models has been developed for predicting the behavior of deepwater oil spills and their impact on the Black Sea marine environment and coasts. In this regard, here, a proposed coupled circulation/oil spill model was elaborated especially to address the transport and behavior of the 3D structure of a rising oil plume resulting from a representative hypothetical deepwater oil blowout in the Black Sea.

The present work is focused on 3D structure and evolution of a hypothetical deepwater oil spill, taking into account the formation of the plume during its rising from a hypothetical wellhead and subsequent spreading on the sea surface. The paper is also addresses contamination of the marine environment and beaches of the Black Sea resulting from the plume development transport. In deepwaters, the movement and fate of a multiphase plume is governed by the gas–oil separation process, rising velocity as well as background currents and stratification, while, at the subsurface layer, the plume evolution and its fate experiences the influence of currents induced by local winds, Stokes drift and physicochemical processes which change the oil properties. Note that the gas–oil separation process is not considered in the paper, it is substituted by a simplified parameterization. A special attention is paid to effects of mesoscale structures of the Black Sea in synergy with basin-scale circulation on the spreading of oil pollution.

## REGION OF INTEREST

The study is focused on the region adjoining the northern Caucasian and Crimea coasts where potential pollution of vulnerable beaches might happen as result of the exploratory deepwater oil drilling in the area over the northwestern edge of the Shatsky Ridge (Fig. 1). The structure of the Black Sea circulation, in this area, is very complex and contains major elements from basin-scales to mesoscales and even sub-mesoscale structures. Figure 2 presents a composite pattern of surface circulation structure of the Black Sea based on historical measurements and satellite data (*Oguz et al., 1993*; *Krivosheya et al., 2001*). Basically, the structure is composed of a seasonally varying (stronger in winter and weaker in summer) the cyclonic basin-scale RC surrounded by numerous near-shore anticyclonic eddies (NAEs) trapped between the RC and the continental shelf. Direct observations based on surface buoys (*Zhurbas et al., 2004*; *Poulain et al., 2005*) and Acoustic Doppler Velocity Profiler (ADCP) measurements (*Oguz & Besiktepe, 1999*) in the upper 100 m have obtained current velocity of about 0.4–0.5 m s$^{-1}$, occasionally increasing up to 1.0 m s$^{-1}$ along the axis of the RC jet. As to deep and intermediate layers, deeper 250 m, i.e., under the RC, as was revealed from drifter tracks (*Zatsepin et al., 2003*; *Korotaev, Oguz & Riser, 2006*) and ADCP measurements (*Ostrovskii et al., 2013*), the current velocity gradually decreased reaching 0.04–0.02 m s$^{-1}$ near the bottom.

A use of autonomous floats allowed providing to *Korotaev, Oguz & Riser (2006)* a clear evidence for well-pronounced currents and an organized flow structure at intermediate (750 m) and deep (1,550 m) layers. It was discovered in contrast to prior assumptions of a rather weak deep circulation of the Black Sea. As the observations also showed, the magnitudes of intermediate and deep currents reached as much as 0.05 m s$^{-1}$ at 1,550 m.

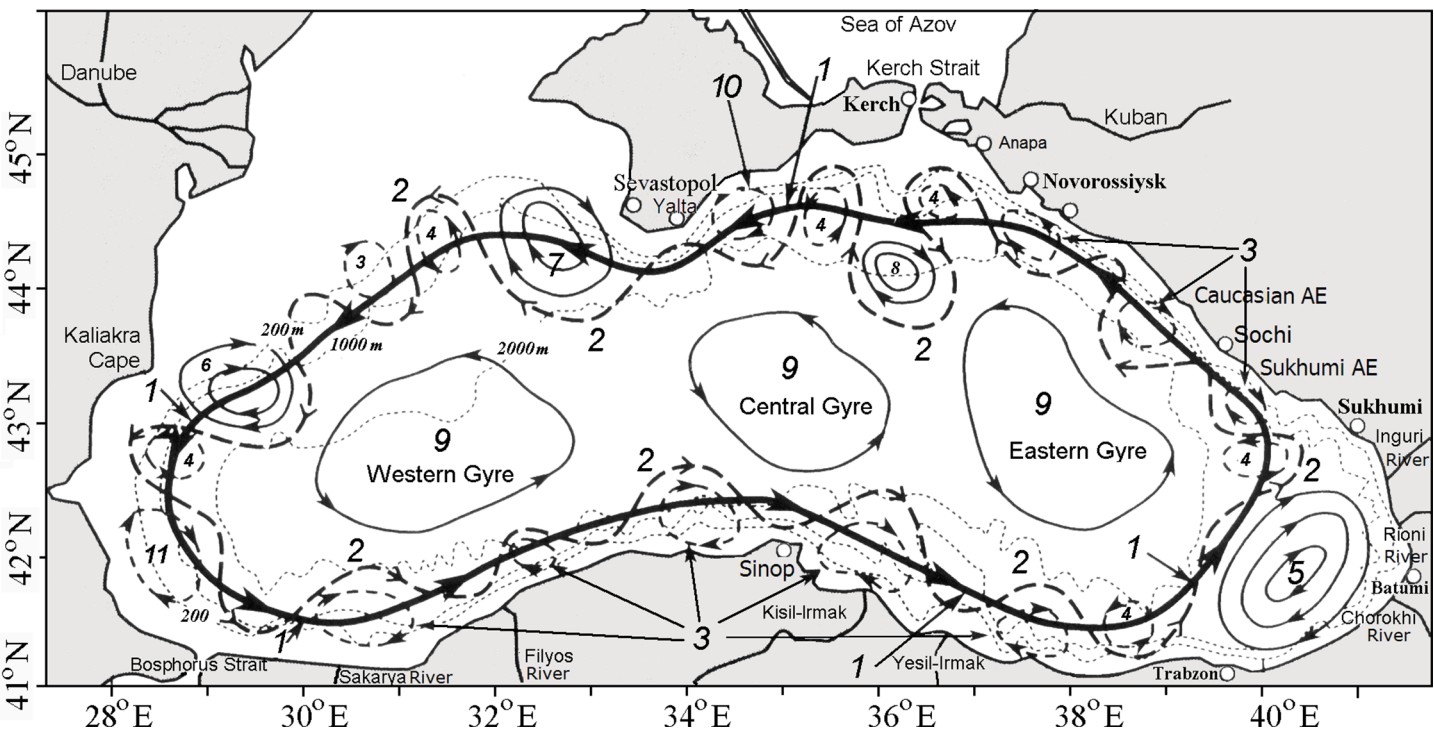

**Figure 2 Schematic of the Black Sea circulation (modified from *Korotenko (2016)*).** 1—mean position of the Rim Current jet; 2—meanders; 3—near-shore anticyclonic eddies (NAEs); 4—cyclonic eddies (CEs); 5—Batumi anticyclonic eddy; 6—Kaliakra anticyclonic eddy; 7—Sevastopol anticyclonic eddy; 8—Kerch anticyclonic eddy; 9—quasi-stationary cyclonic gyres; and 10—Crimea anticyclonic eddy; and 11—Bosphorus anticyclonic eddy.

Note that not all structures presented in Fig. 2 occur at the same time; nevertheless they are very persistent features of the Black Sea (BS). Observations showed that the NAEs have a preponderance to form during summer and autumn when the RC became weak and unstable. Typically, NAEs have diameter ~30–60 km and their shape is close to circular. After their formation due to headland eddy shedding or from baroclinic instabilities, most NAEs remain trapped between the coast and the RC, and "roll" along the coast like "lubrication eddies" in the same direction as the RC (*Korotenko, Bowman & Dietrich, 2010*).

The Caucasian near-shore anticyclonic eddies (CNAEs) periodically appear in the area between Sukhumi and Sochi, preferentially in winter–spring months. Their average lifetime ranges from 2 to 3 months. Moving northwestward along the Caucasian coast, the CNAE is often accompanied by a large offshore anticyclonic meander, the RC being shifted into the central part of the eastern basin. The CNAE often interacts with the Kerch anticyclonic eddy, which is also a well-pronounced element of the Black Sea eddy dynamics. An average persistence of this eddy is about 240 days and its mean lifetime as about 80 days (*Korotaev et al., 2003*). The spring and autumn seasons are revealed to be more favored periods for the presence of the Kerch eddy. The Crimea anticyclonic eddy usually occurs in August and September, its lifetime is about a month.

The Sevastopol anticyclonic eddy (SAE) is also among intense and persistent eddies in the Black Sea. The SAE is periodically formed southwest of the Crimea Peninsula by

intense vorticity generation over the very steep continental margin at the southern tip of the peninsula. As was observed, winter and summer are most preferred periods for the SAE formation. According to satellite and instrumental observations, the largest SAEs can grow up to 100–150 km in diameter and its thickness reaches 100–200 m.

## METHOD

### Deepsea oil spill model

Accurate predicting the transport, dispersal, and fate of oil plume releasing by a deepwater blowout, description the plume rising toward the sea surface and spreading in the subsurface layer require developing new models with ability to describe major processes affecting oil plume and utilizing field observations obtained during and after an accident. For modeling deepwater oil spills, the Lagrangian particle-tracking method (LPTM) coupled with the low dissipative eddy resolved DieCAST ocean circulation model (*Dietrich et al., 1997*) was adapted for the Black Sea, the latter being the $(1/30)°$ horizontal resolution version of the DieCAST (Die2BS) (*Korotenko, Bowman & Dietrich, 2010*; *Korotenko, 2015*).

The overall structure of the deepsea oil spill model (DOSM) is presented in Fig. 3, where oil properties and are geographic data stored in the database in advance. Once a deepwater oil spill accident happens, data such as duration of the spill and its location, oil volume/discharge-rate as well as information on current weather conditions and waves are input into the model. Die2BS has operated ahead of the oil transport model in order to provide the DOSM with necessary hydrodynamic data and parameters. Taking into account a combined effect of winds and waves on the horizontal transport of oil, the superposition of wind and wave drift currents is calculated.

Generally, the procedure predicting oil plume behavior is divided into two parts: (i) pre-calculation of currents, $\vec{V}$, temperature, $T$, and salinity, $S$, and diffusion coefficients, $K_H$ and $K_Z$ with a use of the Die2BS hydrodynamic model; (ii) applying computed mean $\vec{V}$, $T$, $S$, $K_H$, and $K_Z$ and calculated terminal velocity (see below) for each oil droplet to describe movement of individual oil droplets, the sum of which constitutes the oil plume; and (iii) simulating processes of rising oil droplets from the deepwater wellhead, their advection, diffusion, dispersion, biodegradation, and dissolution in the water.

Algorithms for oil evaporation and decomposition (due to biochemical and physical degradation) are incorporated in a special ancillary module, which compares the current model time with the "half-life" time assigned a priory to each droplet (*Korotenko, Bowman & Dietrich, 2010*; *Korotenko, 2016*). A droplet is considered as lost if the current time exceeds "half-life" time assigned to the droplet. Note that only those droplets that occurred within the subsurface "evaporation layer" ($z_{ev} \sim 0.1$ m) experience decay due to evaporation, while disintegration and dispersion may effect on all droplets occurred below $z_{ev}$. In other words, weathering module, shown in Fig. 3, operates only at or near to the surface while processes of dissolution, sedimentation, diffusion, etc., are calculated throughout the water column.

In the DOSM, oil is presented as a mixture of eight hydrocarbon groups (*Mackay & McAuliffe, 1988*) that allows determining evaporation process more accurately. The

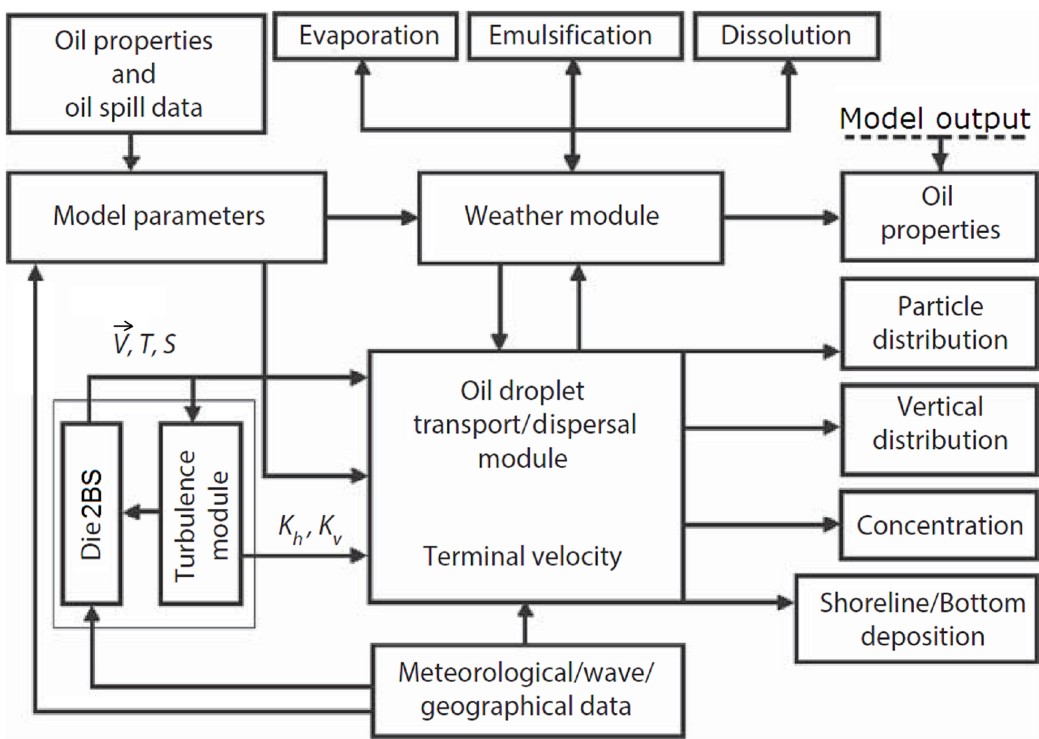

**Figure 3 Schematic of principal elements of the deepwater oil spill model (modified from *Korotenko (2016)*).** $\vec{V}$, $T$, $S$, $K_h$, and $K_v$ denote mean velocity, temperature, salinity, horizontal and vertical viscosities, respectively.

hydrocarbon groups $C_1$–$C_8$ were chosen for light crude oil and adapted for the type oil with characteristics akin to the Black Sea oil. The coupled model was described in detail by *Korotenko (2016)*, below the meaning of main elements and some important equations of the model will be only shortly clarified.

### The behavior of a deepsea oil spill in the marine environment

An important issue for deepwater oil spill modeling is to calculate so-called terminal velocity for each oil droplet, i.e., rise velocity of oil droplets which depends on their properties. The terminal velocity, $w_t$, of an oil droplet is estimated in the oil droplet block (Fig. 3) with a use of the equation for a solid particle (*Perry & Green, 1984*):

$$w_t = \left[ 4gD_p(\rho_p - \rho)/(3\rho C_D) \right]^{1/2} \tag{1}$$

where $\rho_p$ is the density of oil, $\rho$ is the density of water, $g$ is the gravitational constant, $D_p$ is the diameter of a droplet, and $C_D$ is drag coefficient. $C_D$ is determined to be equal to $24/Re_p$ when $Re_p < 0.1$ or equal to $\left(\frac{24}{Re_p}\right)\left(1 + 0.14Re_p^{0.7}\right)$ when $0.1 < Re_p < 1{,}000$. Here, $Re_p = D_p\rho_p w_t/\mu$ where $\mu$ is the dynamic viscosity. It is apparent from Eq. (1), the more size of an oil droplet and greater density difference between oil and ambient water the more terminal velocity of the droplet.

Moving upward with different terminal velocity, the assembly of oil droplets constitutes an ascending subsea plume, in which different droplet will reach the sea surface at different

times after the release. Moreover, due to vertical inhomogeneity of horizontal crossflow droplets with different sizes will appear at the sea surface at different locations. Therefore knowledge of the droplet size distribution in a rising oil plume is very important for adequate predicting plume behavior. In other words, determined by the size of droplets and difference between oil and ambient water, terminal velocity of each droplet will fundamentally control when and where the droplets reach the surface and form the surface slick. In practice, various methods are used to estimate the diameters of oil droplets emanating from a blowout, e.g., they can be measured in a laboratory (*Masutani & Adams, 2001*) or field (*Johansen, 2003*) experiments as well as simulated theoretically (*Chen & Yapa, 2007*).

In the model, the size diameter distribution resembling lognormal distribution which characterized the influence of the natural dispersion on oil transformation was used. Such distribution was obtained in experiments with light crude oil in a wave tank by *Li et al. (2008)* and contains a very small amount of small droplets ranged from 5 to 80 μm and from 500 μm to 1 mm while the vast majority of volume fractions contains droplets ranged from 90 to 400 μm with the median value of 300 μm. For Reynolds numbers picked within the intermediate regime within the range $0.1 < \mathrm{Re}_p < 1{,}000$, the largest oil droplets with a diameter of about one mm will rise with a velocity ranged from 0.0031 to 0.074 m s$^{-1}$, respectively. It means that in the case when the oil well located at the depth of 1,053 m (see below) the droplets will take 93.5 h to reach the sea surface with the former velocity while with the latter velocity it will take only 4 h (cf. *Dasanayaka & Yapa, 2009*; *Lardner & Zodiatis, 2017*). The terminal velocity of oil droplet, from the interval from 90 to 400 μm (for $\mathrm{Re}_p = 1$), ranges from 0.0028 to 0.0059 m s$^{-1}$, respectively. For the calculations, oil with the average density of 830 kg m$^{-3}$ (API = 39) and seawater at 1,017 kg m$^{-3}$ were used.

It should be emphasized that there are significant differences between processes govern surface oil plumes and those govern plumes resulted from deepwater (depth >300 m) blowouts. Summarizing the observations of deepwater oil/gas plumes in crossflow and stratified water *Socolofsky & Adams (2005)* suggested that the following progression of deepwater plume stages:

1. In the proximity of a deepwater oil source, the oil/gas mixture progresses as a coherent plume.

2. Higher, despite the oil/gas mixture is still behaving as a coherent plume; however, a leakage of fluid entrained from the downstream side of the plume begins because the stripping current velocity overcame the restoring entrainment velocity decreasing with height.

3. Next, above a so-called critical separation height, $h_S$, where gas and oil separated, the oil/gas mixture loses its coherency so that entrained water and fine oil droplets are lost downstream while gas and large oil droplets are lost upstream.

4. Finally, the separated mixture of entrained fluid and oil droplets continued to rise in the far-field as a buoyant jet and can be modeled as a single-phase oil plume, initiated at the separation height, $h_S$.
Knowing the critical separation height, one can estimate so-called transient time, i.e., the time when gas escaped from the mixed plume. According to observations and model experiments (*Johansen, 2003*; *Zheng, Yapa & Chen, 2003*; *Chen & Yapa, 2007*; *Yapa & Zheng, 1997*; *Zheng & Yapa, 1998*), the transient time was estimated to be equal to several seconds, so that after time of the transition, the LPTM can be used to describe the transport and dispersal of oil droplets in the far-field plume.

### LPTM algorithm used in DOSM

Once oil droplets rising from a deepwater oil well appear at the surface, they are moving away from initial surfacing points due to the complex action of surface currents, winds, and waves. To predict the movement of an ensemble of oil droplets, in the model, the displacements of each droplet can be estimated as (*Korotenko, 2016*):

$$(\Delta x_i)_{j,k} = V_{i,j}\Delta t_j + (\vartheta_i)_{j,k}$$
$$\left(i = 1-3; j = 1, 2, \ldots, N_t; k = k_f = 1, 2, \ldots, N_f; f = 1, 2, \ldots, 8\right) \tag{2}$$

where $N_f$ is the number of droplets within $f$-th hydrocarbon group while $k_f$ denotes the $k_f$-th droplet within a $f$-th C-group. Hereafter for brevity, subscript $f$ was omitted. The displacements $(\Delta x_i)_{i,k}$ are determined as a sum of a deterministic part of the droplet displacement due to the mean velocity field, $V_{i,j}$ and a random displacement, $(\vartheta_i)_{i,k}$ due to velocity fluctuations determined the block "$\vec{V}, T, S$" (Fig. 3). The term $(\Delta x_i)_{j,k}$ is the displacement of the $k$-th droplet along the axis $x_i$ at the $j$-th instant of time. $N_t$ denotes the total number of time steps, and $\Delta t$ is the time step. The distribution of the number of particles in $f$ groups depends on the type of oil; it is initially assigned and distributed randomly according to the specification of oil chosen. Each $k$-droplet within a $f$-th group is characterized by size, density, position $X_{i,j,k}$ and its "half-life" period. The latter, as was said above, was assigned a priori once the droplet was launched.

The advective movement within a grid cell is computed with the use of the linear interpolation of the velocity components at a droplet position from eight nodes of a corresponding Die2BS grid cell at the time step $\Delta t$.

To estimate random displacements of each droplet due to sub-grid fluctuations of velocity or, shortly, diffusive jumps of a droplet, $(\vartheta_i)_{j,k}$, different approaches for the horizontal ($i = 1, 2$) and vertical ($i = 3$) axes were used. For the horizontal axes, so-called "naive random walk" scheme is widely used. In this approach $(\vartheta_i)_{j,k}$ is defined as $\vartheta_{i=}\gamma_i\left(2K_{i,j}\Delta t\right)^{1/2}$ (*Spaulding, 1988*; *Korotenko, Mamedov & Mooers, 2001*). Here, $\vartheta_i$ is a random vector, normally distributed with an averaged value of 0 and unit standard deviation.

To avoid artificial droplet accumulation in layers with weak vertical mixing, for the vertical axis, so-called "consistent random walk" (CRW) approach is applied. The latter approach was developed by *Visser (1997)* who suggested the following formula for estimating vertical droplet's displacement:

$$\vartheta_3 = K'_3(z)\Delta t + \gamma_3[2K_3(z^*)]^{1/2} \tag{3}$$

The CRW approach describes deterministic and diffusive components of vertical displacements. The deterministic component describes a net displacement of the center of

mass of droplets toward increasing diffusivity expressed by a local gradient of $K_3$, i.e., $K_3'$. It allows avoiding an artificial accumulation of droplets within layers where vertical diffusivity is low. The vertical diffusivity, $K_3$, in the CRW model, is estimated with the use of the diffusivity profile at a vertical coordinate $z^*$ shifted from the droplet coordinate $z$ by a small distance $0.5\,K_3'(z)\Delta t$. More details on LPTM algorithms were given in (Korotenko, 2016). Note, however, that this kind of approach is very sensitive to the vertical resolution of the model; for coarse resolution, the effect of the CRW approach on the accumulation of particles/oil droplets would hardly be noticeable. Nevertheless, for the Black Sea, the generalized oil spill model with the CRW approach was used for a future implementation of a fine resolution nested model.

### Modeling the deepwater oil blowout

A hypothetical oil source was set at the bottom, at the site south of the Kerch Strait over the northwestern edge of the Shatsky Ridge at coordinates 44°33′N, 36°36′E where depth is 1,053 m. In Fig. 1, the source was marked by the asterisk inside of the red dot denoted the discovered oil-rich site (Egorov et al., 2003). The blowout lasted 20 days and its discharge rate was set to be constant and equal to 20 metric tons h$^{-1}$.

Since a plume released from deepwater oil blowout presents, as was pointed above, the oil–gas mixture, the latter will split, at some separation height, $h_S$, above the bottom, into individual oil droplets and gas bubbles. For an instance, the separation height is equal to 200 m for bottom current velocity of 0.02 m s$^{-1}$ (Socolofsky, Adams & Sherwood, 2011), so that, within the 200 m layer above the bottom, oil droplets are driven by gas bubbles while above $h_S$, oil droplets rise with the terminal velocity determined by Eq. (1).

The continuous source of oil droplets was mimicked by regular ejection (its period coincides with the oil spill model time step, $\Delta t$) of a cloud of droplets. In the oil spill model, 1,000 oil droplets were released every 30 min at the depth of hypothetical blowout. Each droplet represents a fraction of the mass of released oil, so that each oil droplet will represent one kg of oil. It gives the initial concentration of oil within the first $z$-layer above the oil wellhead of $1.5 \cdot 10^{-7}$ kg m$^{-3}$. Note that at the time of each ejection of a droplet cloud, a full set of parameters and properties determining the state of each droplet are assigned (Korotenko et al., 2004).

### Oil parameters setup

Simulating the transport and fate of an oil spill requires a specification of a number of initial parameters. Light crude oil, used in the present work, was chosen to be characterized by the following parameters: oil droplet diameters assigned randomly between $d_{min} = 2.5$ μm and $d_{max} = 400$ μm; "half-life times" were chosen as Tev$_1$ = 20 h, Tev$_3$ = 30 h, and Tev$_5$ = 10 h for the hydrocarbon groups $C_5$, $C_1$, and $C_3$, respectively. For the "long-living" groups, $C_2$, $C_4$, $C_6$, $C_7$, and $C_8$, They all were set to be equal to Tev$_4$ = 250 h (Korotenko, Bowman & Dietrich, 2010).

For the chosen oil, the percentage mass ratio between $f$-groups was set as follows: $C_1$ = 15%; $C_2$ = 20%; $C_3$ = 25%; $C_4$ = 10%; $C_5$ = 15%; $C_6$ = 3%; $C_7$ = 7%, and $C_8$ = 5%. Such ratio means that after oil reaches the sea surface it starts evaporate, with 55% of total

oil mass is expected to be evaporated within a first few days. The evaporation will occur mostly due to light hydrocarbon groups $C_5$, $C_1$, and $C_3$.

### Boundary conditions: interaction of oil droplets with shoreline and bottom

The DOSM takes into account the beaching and depositing of oil droplets. In the case when an oil droplet reaches the coastline/bottom, the droplet is considered as beached/deposited one. Vertical boundaries, i.e., the sea surface and bottom are specified by interpolating sea surface level ($z = 0$) and the bottom to the $x$–$y$ location of each oil droplet. There are two types of vertical boundary conditions are used:

1. When a moving droplet passes through the surface or bottom boundaries due to vertical movement then the droplet is returned back to the model domain at a distance equals to the displacement that the droplet exceeds the boundary (the reflecting boundary conditions).
2. When a moving droplet numerically jumps over the surface or bottom then the droplet is returned back to the nearest point of the correspondent boundary and coordinates of the droplet are fixed (absorbing boundary conditions).

The horizontal boundary condition is a reflecting one if routines keep droplets inside the model domain. If the droplet is on land, the droplet is reflected off the boundary. The integration time step is chosen on condition that a droplet remains within a correspondent cell. The horizontal boundary condition routine allows droplets to reflect repeatedly within a time step.

Based on the abovementioned boundary conditions, in the model, special algorithms are used to define a number of oil droplets beached and deposited. Should a droplet reach the coastline or bottom, it is marked as deposited or beached; its coordinates are fixed at the point where the droplet reached the correspondent boundary. This procedure also accounts for the redistribution of the total oil mass between different oil fractions, i.e., oil evaporated, oil beached, and oil deposited. This redistribution of oil is very important for assessing risks and scales of coastline contamination (*Korotenko, Bowman & Dietrich, 2010*).

### DieCAST circulation model

The principal element of the DOSM is the high-resolution, low dissipative hydrodynamic model Die2BS, shown in Fig. 3 and described in (*Korotenko, Bowman & Dietrich, 2010*; *Korotenko, 2015*, *2017*). The computational grid of the model covers the entire Black Sea basin from 27.2° to 42°E and from 40.9° to 46.6°N, and contains a total of 426 × 238 rectangular cells, with 30 unevenly spaced levels in the vertical. In the model, the ratio of the horizontal cell dimensions ($\Delta X/\Delta Y$) is fixed and equal to unity so that square cell dimensions varied only in latitude from 2.6 to 2.8 km. Since the grid size of Die2BS is significantly less than the first internal baroclinic deformation radius $R$ ~5–20 km for the Black Sea, the model is able to adequately resolve near-shore mesoscale structures and their variability.

The Die2BS was initialized with monthly-averaged temperature and salinity data and forced with climatological surface buoyancy (heat) fluxes, evaporation minus precipitation,
monthly winds, and river runoff from 31 rivers (*Jaoshvily, 2002*). At two open boundaries, the exchange through the Bosporus and Kerch Straits are specified as in (*Korotenko, 2015*). Upon the run of Die2BS, a special nudging data assimilation procedure was launched. In doing so, the surface buoyancy flux was computed by nudging both the temperature and the salinity toward monthly climatology as in *Staneva et al. (2001)*.

In the Die2BS, 30 unevenly *z*-levels were spaced with smaller intervals near the surface for better representation of surface processes, which is crucial for oil spill modeling. These levels are distributed as following: 0, 3, 6, 10, 14, 21, 26, 32, 39, 46, 56, 66, 79, 94, 112, 133, 159, 190, 227, 298, 359, 432, 521, 692, 837, 1,014, 1,230, 1,493, 1,645 and 2,221 m. In the model, an unsmoothed *ETOPO2* bottom topography is used, and bathymetry is represented as series of steps, where the vertical velocity is set to 0. The integration time step was chosen to be equal to 6 min. The Die2BS was spun up from rest and with the climatological temperature and salinity. The model was run for a total 23 years with perpetual seasonal forcing, to ensure that the basin averaged kinetic energy, temperature, and basin-scale circulation reach quasi-stationary periodical states. The climatological data used in the DieCAST model has been provided from the CoMSBlack surveys (http://sfp1.ims.metu.edu.tr/texts/database.htm) in the context of the NATO Black Sea project (*Staneva et al., 2001*).

The validation of the circulation model was conducted on the base of satellite images of sea surface temperature and altimetry data, surface current velocities obtained in observations and derived from drifter experiments. Results of the validation were presented in *Korotenko (2015*, *2016*, *2017*).

As was shown in *Korotenko (2015*, *2017*), the Die2BS model realistically reproduces basin-scale circulation of the Black Sea as cyclonic gyres, the quasi-permanent cyclonic RC, and its seasonal fluctuations; Rossby waves propagating westward across the basin; mesoscale structures as eddies, filaments, up- and downwelling events, mushroom currents, jets (*Korotenko, 2015*, *2017*). High resolution and extremely low horizontal dissipation (horizontal viscosity ranging from 5 to 10 $m^2\ s^{-1}$) allow the model to reproduce also numerous anticyclonic eddies and meanders lying between the coast and the RC that is very important from the point of view the transport of contaminations on the continental shelf of the Black Sea.

# RESULTS

## Simulation of the Black Sea circulation

### Mesoscale features along the Caucasian and Crimean coasts

As simulations revealed, a part of the RC jet flowing along the Caucasian coast is periodically displaced offshore. It creates, in the region from Sukhumi to Novorossiysk, large amplitude meanders with favorable conditions for the generation of anticyclonic eddies (Fig. 2). Being a quite persistent structure of this region, the Caucasian near-shore eddies, as was found, often associates with large offshore protrusions of the RC toward the open sea with periodic detachments of the protruded eddies and their absorption by the interior basin circulation. The system "meander-eddy" are often moved along the

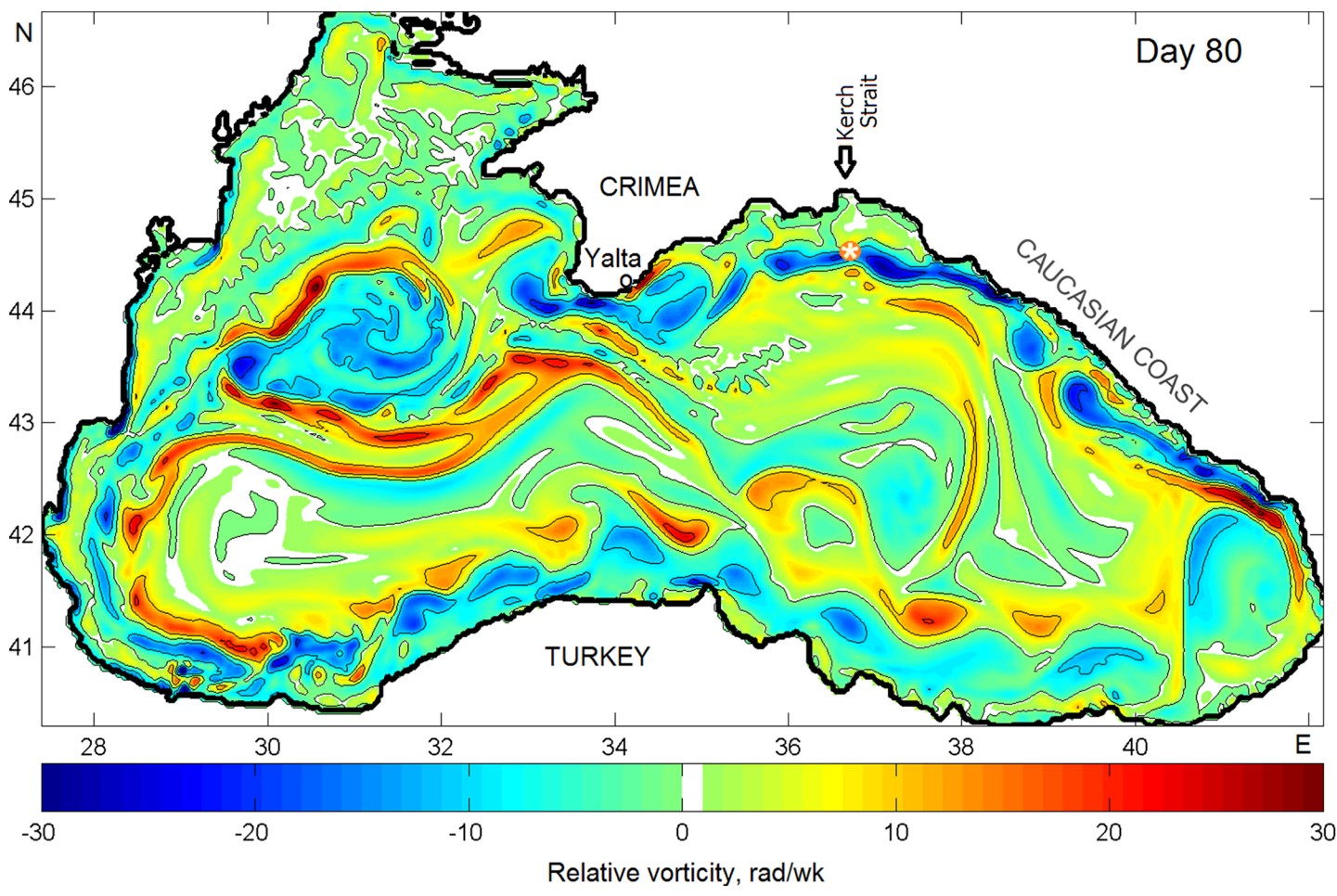

**Figure 4 Snapshots of the relative vorticity in the Black Sea on Julian day 80 of model year 24.** Warm shading denotes positive cyclonic vorticity; cold shading denotes negative anticyclonic vorticity.

Caucasian coast and interacted with the Kerch eddy; the latter is also one of the most pronounced features of the Black Sea eddy dynamics. The Kerch and Caucasian eddies, in turn, could either interact with each other or could be separated from each other by a sharp onshore meander of the RC.

Greatly effecting on the local dynamics, the SAE and Crimean coastal anticyclonic eddy appear asynchronously at the western and eastern sides of the Crimean peninsula, respectively. The Crimean eddy is generally attached to the southern tip of the headland while the location of the SAE depending on the local structure of the RC moves with the latter away from the Crimea, mainly southwestward along the topographic slope zone between the northwestern shelf and the western interior.

To illustrate the structure of eddy activity in the Black Sea, Fig. 4 presents a pattern of the surface relative vorticity, $\vartheta = \frac{\partial V}{\partial x} - \frac{\partial U}{\partial y}$ on Julian day 80 (hereinafter "Day") of model year 24. Here, $U$ and $V$ are mean velocity horizontal components. The relative vorticity characterizes a measure of rotation at any point of the sea. Counterclockwise rotation means positive or cyclonic vorticity (warm shading) while clockwise rotation denotes

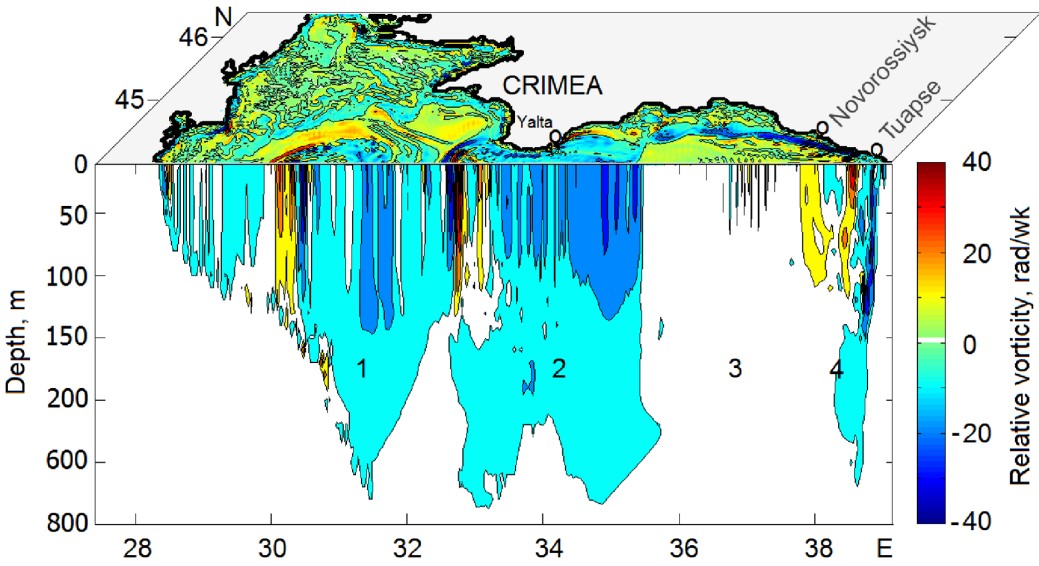

**Figure 5 Composite plot of the surface and vertical relative vorticity for the same day as in Fig. 4.**
The section was made along 44N. Numbers 1–4 are explained in the text.

negative or anticyclonic vorticity (cold shading). The sign of ϑ allows identifying mesoscale eddies arising in the sea and, hence, their effect on spreading of pollutants in the Sea. Anticyclonic eddies create convergence zone and, therefore, may accumulate pollution inside of them while cyclonic eddies, on the contrary, reinforce pollution dispersion due to the divergence of circulation.

Comparing Fig. 4 with the circulation scheme shown in Fig. 2, one can see a chain of anti- and cyclonic eddies embedded into the RC including large SAE southwest of the Crimean peninsula and mesoscale eddies along the Anatolian coast of Turkey. Of interest here is anticyclonic activity along the Caucasian and Crimean coasts. As seen, two Caucasian NAEs appeared, by Day 80, offshore in the central part of the Caucasian coast. Farther northwest, stretching zone of anticyclonic vorticity crosses the location over the oil source and approaches the well-pronounced Crimean NAE west of town of Yalta. According to the numerical simulation, in shallow waters, the thickness of NAEs is limited by depth but do not often exceed 200 m as they move away from the shore.

Figure 5 presents the composite planar plot of the simulated relative vorticity at the surface combined with the vertical section of ϑ made along 44°N. As seen, a core of both anticyclonic and cyclonic eddies embedded in the RC spanned down to about 150 m albeit anticyclonic vorticity penetrates deeper. Light blue color denoting weak negative rotation ($>-20$ rad wk$^{-1}$) extends down to about 700 m while penetration of the cyclonic vorticity is limited by 130–150 m. In Fig. 5, the section passes through four zones which can be identified as follows: (1) zone 1 crossing the Sevastopol eddy delineated by cyclonic vorticity west of it and anticyclonic vorticity east of it; (2) zone 2 crosses south of the Crimea with strong anticyclonic activity originated by the Yalta NAE; (3) stagnant zone 3 with extremely weak rotation is in the eastern Gyre far from the RC and (4) zone 4 where the RC approaching the shore in the region of Novorossiysk–Tuapse creates a

system of eddies with different rotation signs. Here again, anticyclonic eddies are found to penetrate noticeably deeper then cyclonic ones. Such results confirmed by field measurements by *Zatsepin et al. (2008)*. Such difference was explained theoretically by a shorter life of cyclonic eddies due to their more intense radiation of Rossby waves than anticyclonic eddies do (*McDonald, 1998*).

## Modeling the transport of oil plumes released by a deepwater blowout

To investigate a combine impact of the meandering RC and mesoscale eddies on the behavior of an oil plume rising from the bottom, as was mentioned above, the hypothetical deepsea oil source was set at the bottom site (1,053 m) east of Crimea (44°33′N, 36°36′E) and two scenarios with the 20-day oil blowout experiment were considered for model year 24.

### The inner structure of rising oil plume

Before considering the two scenarios chosen, it is worthwhile to scrutinize an initial stage of the oil plume development. Figure 6A illustrates the inner structure of the oil droplet distribution in the rising oil plume formed in a 24 h test experiment performed on Day 80. As seen, the rising oil plume consists of four parts: (1) the lower part of the plume, stretching from the bottom to 200 m above it, presents a narrow column of oil droplets driven by gas bubbles; (2) the second part, stretching from 800 m above the bottom to about 400 m below the sea surface, represents an ensemble of oil droplets driven by buoyancy forces with a manifestation of deflection of oil plume by the crossflow. The thickness of this part of the plume is determined by the ratio between the terminal velocity and crossflow in deep layers. An example of the zonal component of subsea crossflow at the depths 200–1,000 m is depicted in the inset (A1); (3) the third part, stretching from about 400 m below the sea surface to about 50 m where an influence of the RC becomes essential; and (4) the uppermost part of the plume above the 50 m where the oil plume experiences a dominant influence of the RC, schematically denoted by RC-arrows in Fig. 6A.

The crossflow presented in insect A1 of Fig. 6A was measured at 44°28.28′N, 37°56.24′E with ADCP near Port of Novorossiysk (*Ostrovskii et al., 2013*). As seen, the zonal velocity in the water column does not exceed 0.2 m s$^{-1}$. Note that, for clarity, droplet colors denote depth ranges in the plume, i.e., magenta, red, green, and blue correspond to depth ranges 0–250, 250–500, 500–750, and 750–1,000 m, respectively.

An examination of the simulation results revealed that a great amount of small droplets, which accounts for a large fraction of oil, remains underwater because smaller droplets move upward much slower than large ones. It means that the horizontal deflection of the smaller droplets by the crossflow is larger than that of large oil droplets. Note that differences in terminal velocity (Eq. (1)) of oil droplets with different sizes lead to different times of surfacing and, thus, positions of the oil droplets on the sea surface. According to the simulations, the first particle reached the surface in about 4 h after the release and its surfacing position was about 200 m north and 150 m west from the projection of the release point on the sea surface. The small droplets remained underwater for a longer

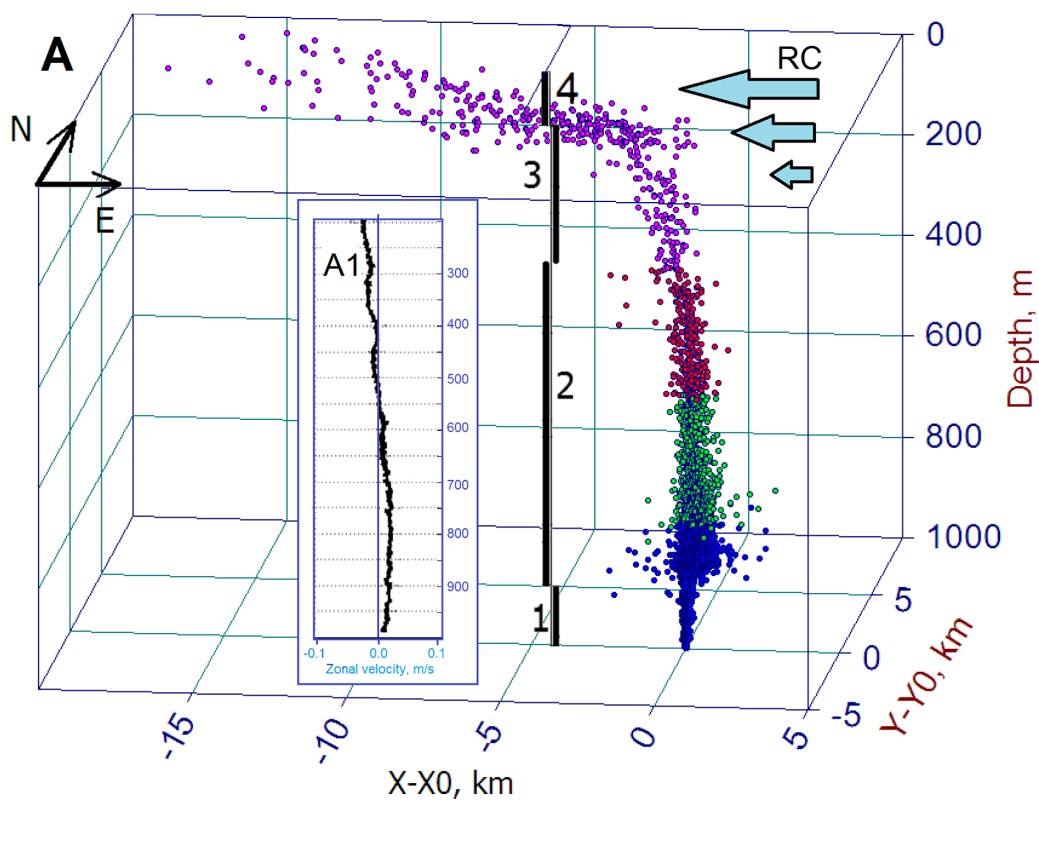

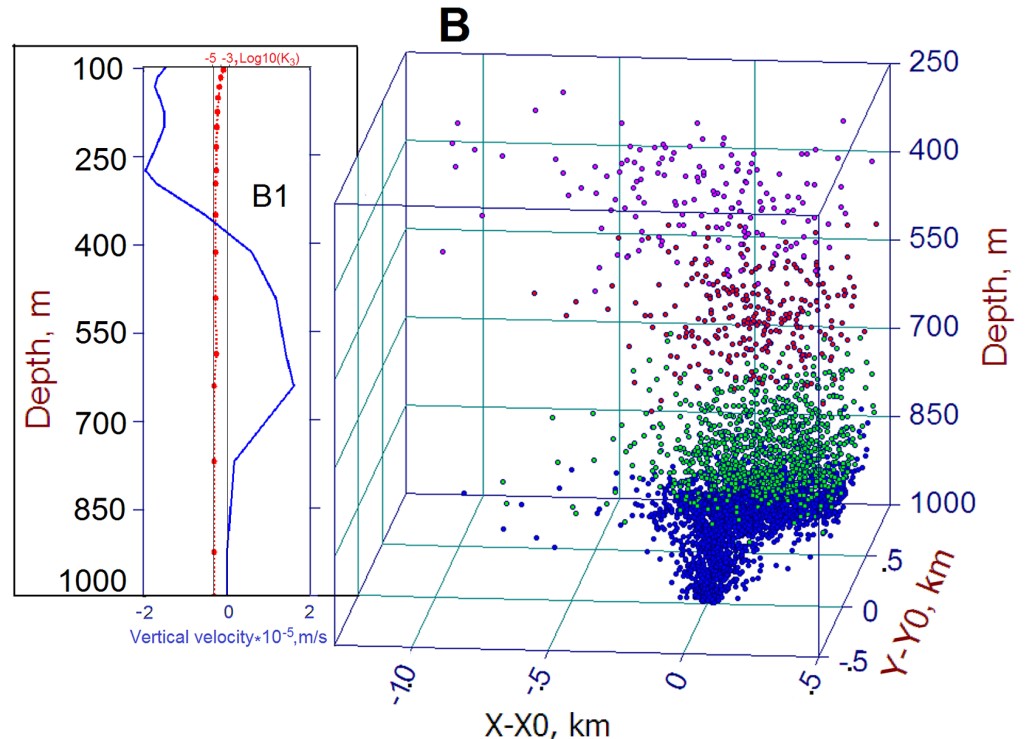

**Figure 6 Inner 3D structures of the initial stage of a deepwater oil plume in the entire water column (A) and in 250–1,000 m layer (B), respectively.** Inset (A1) shows a pattern of zonal velocity in the 200–1,000 m layer (ADCP measurements). Inset (B1). Numbers 1–4, in (A), are explained in the text. Colors of droplets refer to depth range in which they are located: magenta, red, green, and blue correspond to 0–250, 250–500, 500–750 and 750–1,000 m, respectively. Three blue arrows schematically show the Rim Current (RC). N and E indicate north and east directions.

period and, thus, might appear at the sea surface at a larger distance downstream. The horizontal distance that a droplet displaced from the blowout location to the surfacing point is also increasing with time. Figure 6A indicates that for the first day after oil released, the distance that oil droplets traveled on the surface exceeded 15 km while, at the depth below 250 m, the range of oil droplets did not exceed one km.

To highlight an influence of the crossflow on oil droplet displacements, Fig. 6B shows the distribution of droplets in the 250–1,000 m layer where effect of the RC is inessential and, thus, horizontal intrusions of oil droplets are well-pronounced. Following the crossflow (inset A1), within the layer of 800–700 m, most droplets (blue and green color) have a tendency to move eastward while droplets in the layer above 400 m are deflected to the west experiencing an influence of the lower part of the RC. In the intermediate depths from 400 to 700 m, oil droplets (red color) are distributed symmetrically due to negligible crossflow at these depths. Interestingly, despite most oil droplets, within the layer of 800–700 m, move to the east, some of them spread also to the west creating horizontal intrusions of about the 1.1 km-long stretch. It should be also recalled that below this layer oil droplets are driven upward by rising gas bubbles due to the parameterization which artificially mimics this process in the DOSM.

The inset B1 shows profiles of modeled vertical velocity, $V_3(z)$ (blue line), and vertical diffusivity, $K_3(z)$ (red line), above the location of the oil source. As seen, in the 100–1,000 m layer, $V_3(z)$ ranges from $-2 \cdot 10^{-5}$ to $1.8 \cdot 10^{-5}$ m s$^{-1}$ with positive values below 400 m and negative values above 400 m. $K_3(z)$ sharply diminishes with depth from $10^{-3}$ m$^2$ s (at 100 m) to $10^{-5}$ m$^2$ s, so that the noticeable contribution of vertical diffusivity in droplet displacements should be expected in the surface layer of 0–150 m, particularly, in the upper 20 m where vertical diffusivity can play a significant role in the process of natural dispersion (*Dietrich et al., 2014*).

### Scenario 1

In the first 20-day oil blowout scenario, two comparative experiments for investigating combined effects of winds and the RC on the behavior of the oil slick were performed. The oil spill model was run for model year 24 with the blowout starting from Day 20 (i.e., 20 January). In the first run, average climatic winds were used for the Die2BS spin-up while, for the second run, average climatic winds were substituted by daily NCEP winds averaged for years 1998–2002.

Figure 7 shows the final stage of the distributions of oil concentration, log10(C), at the sea surface by Day 40. Figure 7A presents log10(C) in the slick formed under the average climatic winds while Fig. 7B presents that formed under the NCEP winds. As seen,
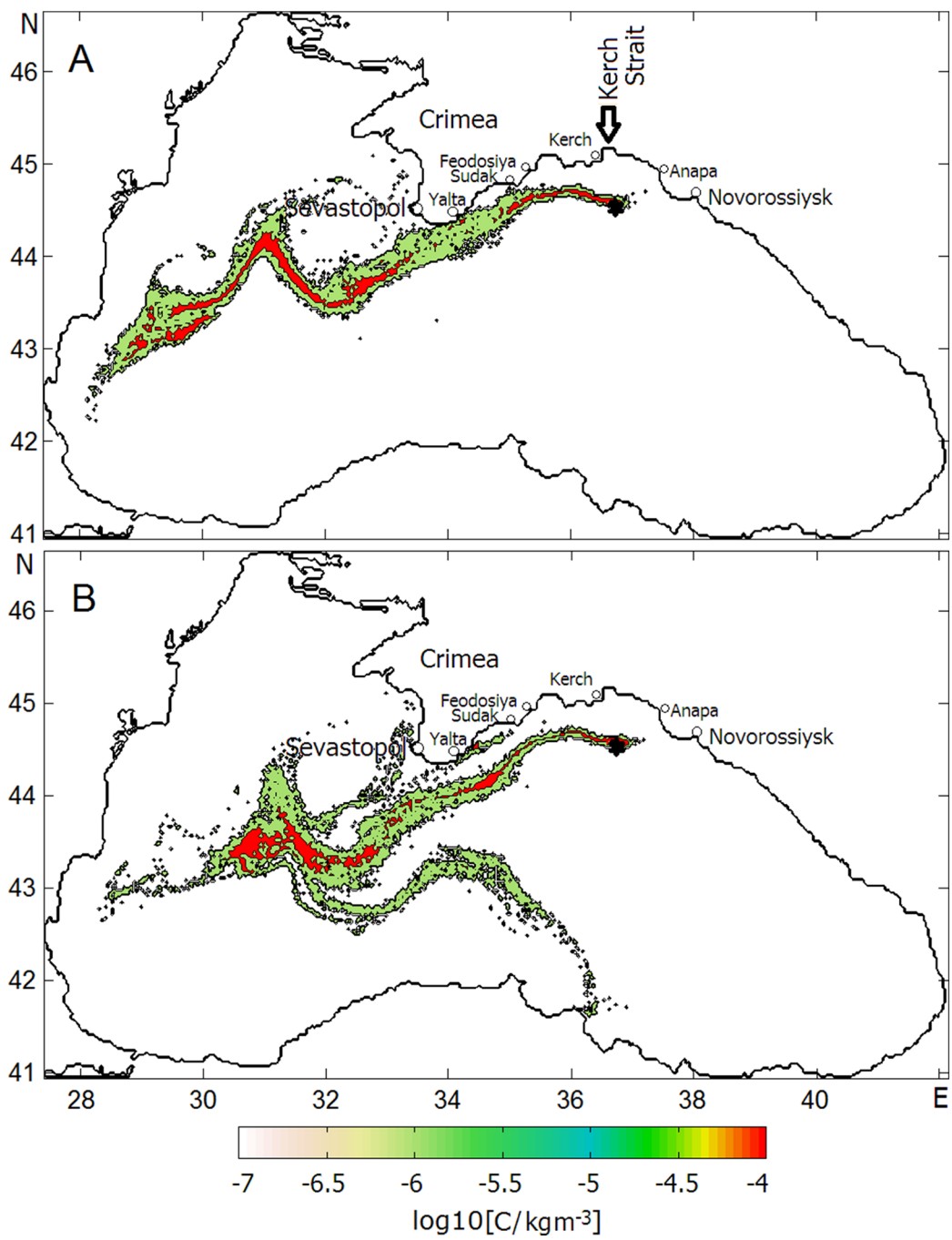

**Figure 7** **Final phases of oil slick development in the 20-day oil blowout experiments under (A) mean climatic winds and (B) daily average NCEP winds.** Sevastopol, Yalta, Sudak, Feodosiya, and Kerch are Crimean towns; Anapa and Novorossiysk are towns at the Caucasian Coast. Arrow indicates the Kerch Strait.

the RC plays the dominant role in the transport of oil at the sea surface, at the same time winds make adjustments to the general distribution of oil droplets in both cases. While under smoothed climatic monthly winds the oil slick following the general direction of the RC flows to the southwest (Fig. 7A), more chaotic daily NCEP winds

spread oil over a wider area so that some amounts of oil are trapped by mesoscale elements of the Black Sea circulation leading to the formation of multiple filaments in the oil slick (Fig. 7B). Interestingly, some filaments caused to split the oil slick and force one branch to move southwest down to Turkey coast. Both experiments indicate that some amount of oil are entrained by the SAE and accumulated along its periphery. As to the contamination of beaches and the bottom, the model predicts, in the first case, only eight and seven tons of beached and deposited oil, respectively, while, in the second case, in the result of stronger wind effect 166 and 176 tons of oil are expected to be on beaches and at the bottom, respectively. In both cases, beached oil contaminated the eastern coast of the Crimea including its southern tip.

### Scenario 2

The next scenario may be referred to as one of the most hazardous events for the marine environment when the oil plume rising toward the sea surface is likely to be captured by coastal eddies approaching the plume. Accumulation of oil droplets inside of the convergence zone created by the anticyclonic eddy and its movement along the coast may present a severe threat not only to the marine environment but also vulnerable beaches. To scrutinize a combined effect of the basin-scale RC, winds, and local nearshore eddies, an often-observed event when the CNAE moving northwestward along the northeastern Caucasian coast to the Crimean peninsula is considered. Understanding behavior of the rising oil droplet plume requires knowledge of its interaction with underwater crossflow and surface current as well as an influence of ambient water density on the spreading of oil droplets. Therefore, first of all, features of local dynamics of the Caucasian and Crimean coastal waters will be considered to elucidate how it may affect the oil plume behavior. For this, pre-described with the Die2BS mean velocity, $V$, temperature, $T$, and salinity, $S$ fields for the period of favorable for the generation of Caucasian eddies (winter) were chosen.

Figure 8 shows sequential snapshots of the simulated sea surface height and streamlines on Days 30, 40, 45, and 50 (Figs. 8A–8D) of year 24. As seen, there is the well-pronounced chain of coastal anticyclonic eddies, including the dominant SAE southwest of Crimean peninsula, embedded in the RC, and the cyclonic vorticity zone in the central part of the Black Sea divided into the western, central and eastern gyres. Figure 8A indicates that, by Day 30, the well-pronounced CNAE has been formed along the Caucasian coast southeast of Novorossiysk.

According to the prehistory, the generation of the CNAE has been triggered by a large offshore protrusion of the RC in the region between Sochi and Sukhumi. As the CNAE was shifting toward the Kerch Strait, it was growing during the following 15 days and then protruded into RC's cyclonic meander. In result of squeezing of the peripheral southern flank of the cyclonic meander and its detachment (light blue shading) by the end of oil experiment (Fig. 8D) a dipole structure of circulation consisting of the CNAE and the cyclonic eddy was formed right over the deepsea oil source. Such evolution of CNAE above the source will strongly affect the behavior of the rising oil plume, surfacing oil droplets and, hence, the transport and transformation of the oil slick.

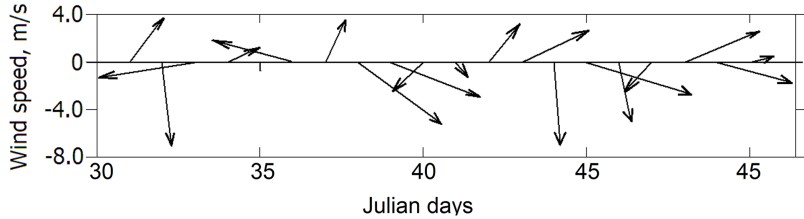

**Figure 8 Sea surface height and streamlines indicating the evolution of the Caucasian anticyclonic meander and CNAEs embedded in the Rim Current.** (A–D) correspond to Julian days: 30, 40, 45, 50, respectively. SAE CNAEs denote the Sevastopol and Caucasian anticyclonic eddies, respectively, while CE is a cyclonic eddy that, along with the CNAE, creates a dipole structure by the Day 50. The black asterisk indicates the position of the deepwater oil source.

**Figure 9 NCEP winds during oil spill experiments.** Each vector indicates the average wind for years 1998–2002.

According to the "Scenario 2," the oil source was launched from Day 30 of simulated year 24. For the period of the experiment, average NCEP winds over the point of oil release are presented in Fig. 9. As seen, moderate unsustainable winds ranging from 1 to 7 m s$^{-1}$ were blowing periodically from the southern and northern directions.
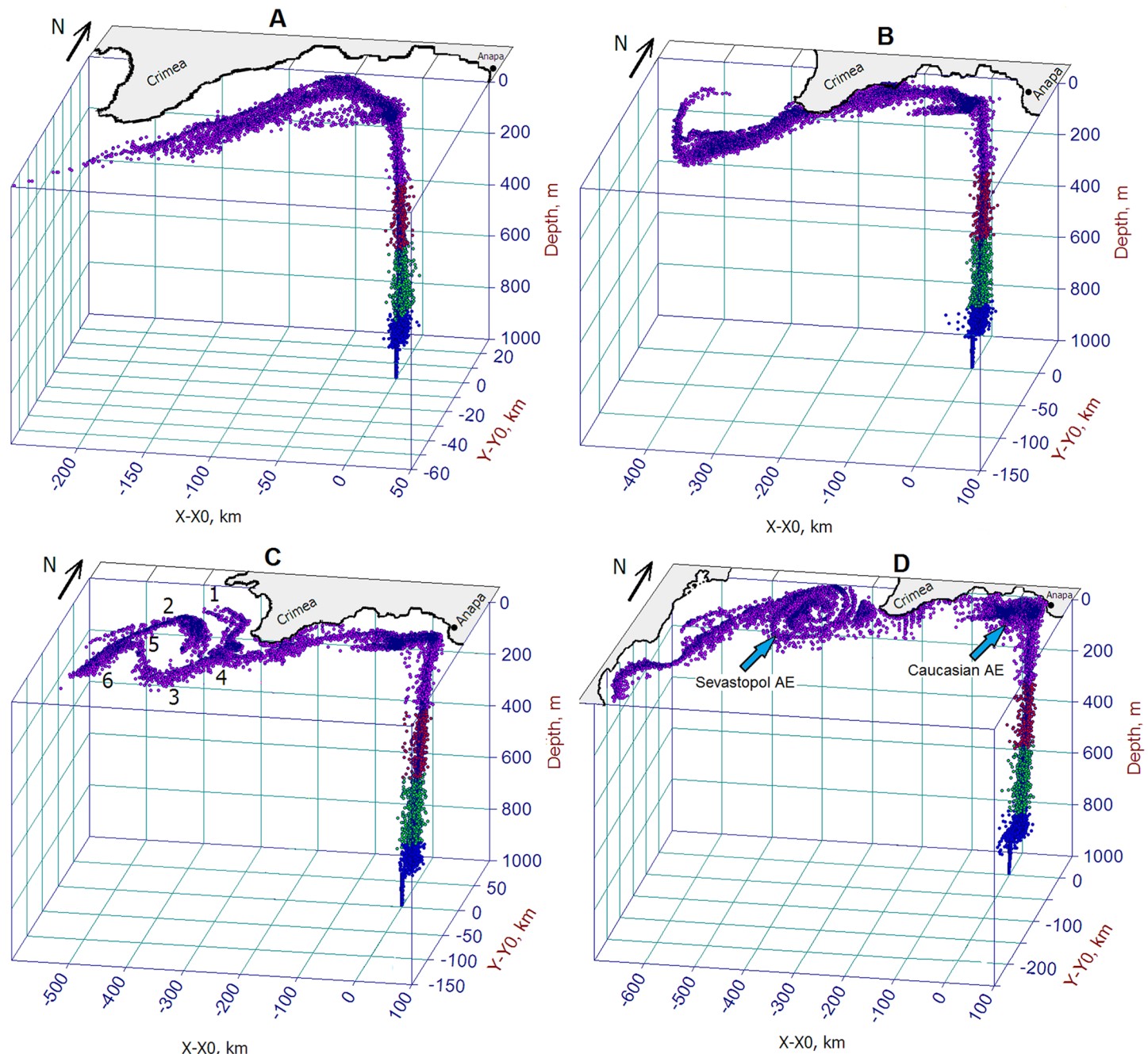

**Figure 10 Successive phases of the oil plume rising from a hypothetic bottom oil source.** Horizontal axes indicate the distance from the oil source with coordinates (*X0, Y0*). Distribution of droplets shows the formation of a 3D oil plume consisting of the underwater part of a rising plume and a surface slick spreading under the actions of the Rim Current and eddies, which captured oil. Panels (A–D) correspond days of the plume development: 35, 40, 45, 50, respectively. Numbers 1–6, in (C), are described in the text. The droplet color corresponds to that as in Fig. 6.

For two initial days of oil experiment (Fig. 10A), the surfaced oil slick moving northwestward and finally approached the Crimean coast despite the competing action of wind drift caused by southwesterly wind and the local loop of the RC directed to the southwest. For the next 2 days, the strong northwesterly wind turned to the northeast

and then, for 2 days, it started to blow from the southwest with speed dropped to about 2 m s$^{-1}$. Such sequence of winds and local circulation has led to the deflection of the oil slick to the southwest and its dragging along the eastern Crimean coast.

The beginning of the following 5-day period of the experiment (Fig. 10B) was characterized by strengthening of southeasterly wind, which later changed to the westsouthwesterly wind of about 4 m s$^{-1}$ and then turned to strong northwesterly winds of about 7 m s$^{-1}$. That sequence of the winds in synergy with the RC action deflected the oil slick to the eastern coast of Crimea. Farther, the oil slick was pressed against the eastern Crimean coast that results in intense oil beaching at the southeastern and southern Crimean coasts. By the end of Day 40, the front-end of the oil slick propagating southwestward was involved in the anticyclonic rotation due to its capture by the SAE.

For the period from Day 40 to 45, winds varied from 3 to 7 m s$^{-1}$ and blew mostly from the north. By Day 45 (Fig. 10C), the sequence of the winds caused the oil slick to move southwestward to the Crimea. As the result of combined effects of the RC, SAE and wind variability, the slick split twice, as was shown by the marks 4 and 5. The branch-1 of the oil slick was originated as a result of the bifurcation of the RC (mark 4) and propagation of oil with cyclonic current flowing along the western coast of the Crimea. The branch-3 was originated by oil flowing with the mainstream of the RC and its deformation by the SAE. After bifurcation (mark 5), the branch-2 was involved in the rotation with the SAE while the branch-6 kept propagating to the south following the RC.

During Days 45–50 (Figs. 10C and 10D), winds kept blowing from the northern sectors and their speed varied from 3 to 8 m s$^{-1}$. As a result of the capture of surfaced oil by the CNAE situated over the point of the oil release, a considerable amount of oil accumulated in the area south of the Kerch Strait and Anapa. The accumulation of oil inside the convergent zone generated by the CNAE led to exhaustion of delivering oil at the eastern coast of the Crimea. Therefore, by the end of the experiment, the oil slick has broken into three main parts: (1) oil captured by the CNAE; (2) remnants of the oil spill stretched along the eastern coast of the Crimean peninsula resulting in beaching and depositing and (3) oil penetrated into and/or trapped by the SAE. Some amount of oil entrained in the RC kept propagating downstream bypassing the SAE. During the entire period of the experiment, the front-end of the oil slick reached 28.5°E, 42.5°N, i.e., south of Kaliakra peninsula (Bulgaria).

### Predicting contamination of shorelines

Generally, the arrival of oil on the shore is the first indication of an offshore oil pollution accident. Depending on the quantity of oil involved, a clean-up response may have to be organized to remove the oil and to prevent it remobilizing and affecting sensitive areas nearby. That is why a reliable early prediction and estimation of the extension of a pollution zone is the important issue in determining the appropriate scale of clean-up operations to be planned. However, the proximity of the offshore exploration drilling site to the Caucasian and Crimean beaches, in case of an accidental oil blowout, allows
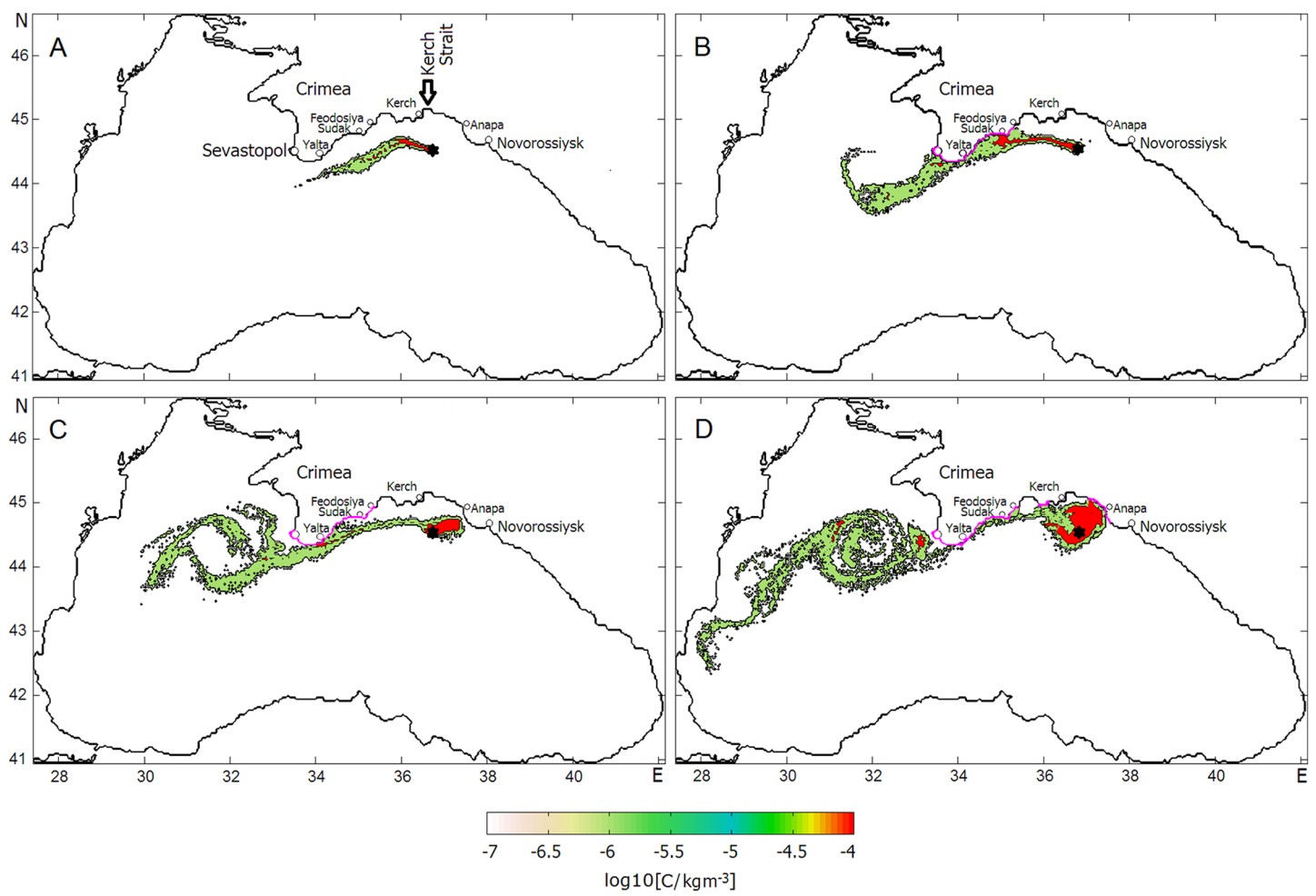

**Figure 11 Successive phases of oil slick development under the influences of the Rim Current, NCEP winds and the Caucasian near-shore anticyclonic eddy.** Magenta color denotes coastlines contaminated by oil. Red color indicates the highest level of oil concentrations in the slick. Panels (A–D) correspond Julian days as in Fig. 10.

only very little time to react and prepare for the clean-up operations. So that any accidental oil spill may cause extreme impacts on the local marine ecosystems and shorelines with negative long-term consequences.

To assess the scale of shoreline contamination, the behavior of the oil slick resulting from the surfacing of the deepwater oil plume in the experiment under "Scenario 2" was examined. The successive phases of the oil slick development and distribution of log10(C) are presented in Fig. 11, in which the coastline contaminated by oil is marked with the magenta color. During the first 5 days (Fig. 11A), the oil slick flowing with the cyclonic RC was spreading southwestward under an influence of winds. There was no indication of oil beaching yet during that period. Within the following 5-day period (Fig. 11B), the meandering RC along with the wind action turned the slick in direction to the Crimean coast that caused the intense beaching and depositing of oil. During that period, the coastline from Feodosiya to Sevastopol was predicted to be completely covered by oil.

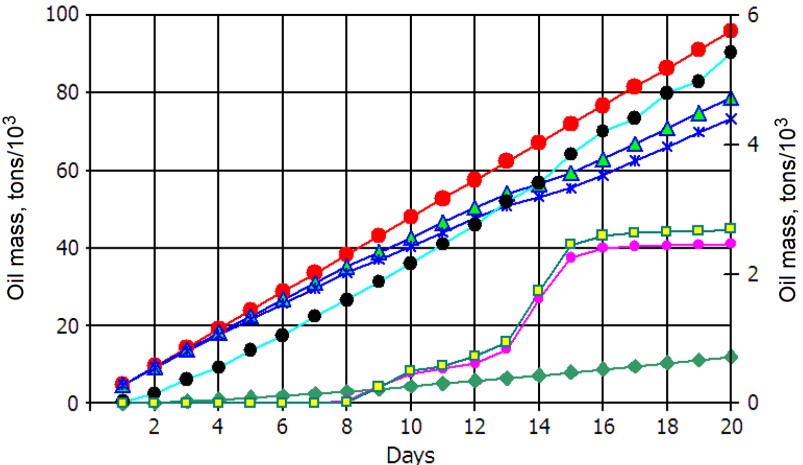

**Figure 12 History of the oil mass balance for scenario 2.** Oil mass components: oil spilled (red dots); dispersed (blue asterisks); evaporated (green diamonds); surfaced (black dots*); total in the water column (light green triangles); deposited on the bottom (yellow squares*); beached (magenta dots*). The components with asterisks refer to values on the right $y$-axis.

In the 40–45 Day period (Fig. 11C), the changing of the wind direction and meandering of the RC turned the spill slightly offshore albeit it still remained to be attached to the southeastern tip of the Crimean peninsula and kept contaminating the shore and bottom.

During the 45–50 Day period (Fig. 11D), the CNAE slowly traveling westward has passed above the site of the oil release causing the capture of oil by the eddy and intense accumulation of oil inside of it. The extension of the oil-contaminated area would result in delivering a large amount of oil onto the shore near the town of Anapa, the well-known summer children's recreation area at the Black Sea coast. Another vulnerable area to be contaminated is the eastern coast of the Crimea between the towns of Kerch and Feodosiya.

This experiment has revealed a key role played by the CNAE in the spreading of oil pollution when it passes above the oil release site. The approaching CNAE intensively captures rising oil droplets due to the convergent circulation created by the anticyclonic near-shore eddy, causes to the isolation of the oil slick core from the open sea and, thus, intensifies further oil accumulation over the blowout site. Such a highly "oiled" eddy being expanded and transported near the coast will massively contaminate the Caucasian and Crimean beaches.

Figure 12 summarizes the oil mass balance as a history of the 20-day continuous oil release. Here, red dots denote total oil spilled, blue asterisks indicate oil mass naturally dispersed throughout the water column, green diamonds show oil evaporated while the oil remained beneath the thin evaporative surface layer ($z_{ev}$) is presented by light green triangles. Mass of surfaced oil is presented by black dots. Amounts of oil deposited on the bottom and discharged onto the coastline (beached) are shown by yellow squares and magenta dots, respectively. Note that last three components of oil balance indicated by asterisks correspond to values shown at the right ordinate. As simulations revealed, the

**Table 1 Change of mass balance components relative to total oil spilled (in % *Wt*).**

| Day | 1 | 2 | 3 | 4 | 5 | 6 | 7 | 8 | 9 | 10 | 11 | 12 | 13 | 14 | 15 | 16 | 17 | 18 | 19 | 20 |
|---|---|---|---|---|---|---|---|---|---|---|---|---|---|---|---|---|---|---|---|---|
| Evaporated | 0.4 | 1.9 | 3.7 | 5.0 | 6.0 | 6.7 | 7.4 | 7.9 | 8.5 | 9.1 | 9.6 | 10.1 | 10.7 | 10.7 | 11.1 | 11.3 | 11.6 | 11.9 | 12.2 | 12.5 |
| Surfaced | 0.5 | 1.5 | 2.5 | 2.8 | 3.4 | 3.6 | 4.0 | 4.1 | 4.3 | 4.5 | 4.6 | 4.7 | 4.6 | 5.0 | 5.35 | 5.4 | 5.4 | 5.5 | 5.4 | 5.6 |
| Dispersed | 99.1 | 96.4 | 93.7 | 92.0 | 90.5 | 89.6 | 88.5 | 87.7 | 85.9 | 84.3 | 83.5 | 82.7 | 79.1 | 79.1 | 77.0 | 76.6 | 76.7 | 76.5 | 76.6 | 76.4 |
| Beached | 0 | 0 | 0 | 0 | 0 | 0 | 0 | 0.1 | 0.6 | 0.9 | 1.0 | 1.0 | 2.4 | 2.4 | 3.1 | 3.1 | 2.9 | 2.8 | 2.6 | 2.5 |
| Deposited | 0 | 0 | 0 | 0 | 0 | 0 | 0 | 0.0 | 0.5 | 1.0 | 1.0 | 1.2 | 1.4 | 2.5 | 3.4 | 3.3 | 3.2 | 3.0 | 2.9 | 2.8 |

contamination of the coastline and bottom began in 8 days after the oil release. Farther, the beached and deposited oil was rapidly increasing by the Day 16, after that the oil mass remained practically unchanged. By the end of the experiment, about 2,100 and 2,200 tons of oil was beached and settled at the bottom, respectively. The mass of evaporated oil reached about 12,000 tons by end of the experiment. As also seen, during the experiment the mass of evaporated oil was growing with the time, reflecting the process of continuous spilling of oil and its rising toward the sea surface, where the light fractions of oil are able to evaporate.

Interestingly, the mass of oil at the surface, i.e., within the model thin sub-surface layer of evaporation ($z_{ev} = 0.1$ m), was only 5,500 tons (5.6%) compared to the total released oil of 96,000 tons. The total mass of oil spreading beneath the sea surface was about 80,000 tons, among which the mass of oil dispersed within the active 50 m surface layer was about 75,000 tons while only 5,000 tons were distributed in the lower column of the plume.

Table 1 lists the mass balance of the spilled oil components over a 20-day period at the indicated time. The table lists the % weight relative to total mass spilled, as is shown by the red curve in Fig. 11. Rising toward the sea water surface, the light component of oil spreads at the surface and light-end hydrocarbons (most volatile components) evaporate to the atmosphere. As seen, the mass of the evaporated oil increases with the time reaching 12.5% at the end of the experiment. As was mentioned above, the evaporation of oil comes out from the thin (0.1 m) layer within which the mass of oil also increases and reaches 4.6% by the Day 20. The mass of dispersed oil, which is conventionally determined as total oil mass distributed from beneath "evaporative" layer to the bottom, decreases with the time down to 79.1% by the end of the experiment. Light oil fractions that reach the shoreline can become stranded ashore (beached). The amount of beached and deposited oil started to grow from the Day 7 and reached, by the Day 15, maximum of 3.1% and 3.3%, respectively. Dispersed oil is considered to be deposited when it reaches the bottom; in the model, this amount is counted separately from the beached oil. Interestingly, after reaching maximum both beached and deposited components of oil mass tend to decrease despite their absolute values, in this period, slightly grows (cf. Fig. 12). It is certain to be associated with the CNAE approaching, which captures oil and detains it inside the eddy.

## DISCUSSION

When considering the results of this study, its limitations should be kept in mind. In particular, this study focused on only two scenarios conducted in winter albeit it is clear

that for different seasons more intriguing results of modeling deepwater oil plumes might be obtained. Moreover, results will also depend significantly on initial conditions. For example, a use of different winds in two experiments under "Scenario 1" led to significant changes in the behavior of the oil slick and modified the final distribution of oil. On the other hand, a use of the same NCEP winds in "Scenarios 1 and 2" but different time of release led also to tremendous dissimilarity in the behavior of the oil slick and different grade of shore contamination. The goal of this study was to illustrate the sensitivity of the coupled model to predict the influence of mesoscale eddies on the transport of oil pollution resulting from a possible accidental deepwater blowout in the Black Sea. It should be emphasized that the present paper focused on methodology, rather than aiming at accurate oil spill predictions, because of large uncertainties in a deepsea oil spill blowout that may happen. Such uncertainties may be eliminated and exact parameters are specified only during the period of a real event. Nevertheless, the model can qualitatively predict what reaction and consequences should be expected in case if CNAEs will approach the oil slick resulting from the deepwater blowout. Besides, the coupled oil spill model allows assessing the oil mass balance and inferring those components related to contamination of the shore and bottom. Estimating bottom contamination is also important as like as the shoreline pollution since due to weathering and dispersion processes some amount of oil may sink and deposit at the bottom. Deposited oil materials, often existing in the form of tar-balls, may be churned up form shallow coastal waters by future storms and blown ashore as was happen when hurricane Isaac (*Dietrich et al., 2014*) passed over the GoM in 2 years after the BP DWH accident in 2010.

## CONCLUSION

In the Black Sea, the advection of NAEs is regarded as one of the most effective mechanisms of horizontal water/pollution transport and exchange between coastal zone and the open sea, and, thus, can be considered as the key mechanism of self-cleaning the coastal zone of the sea. However, such a mechanism can work effectively only in cases when the RC is unstable and nonpersistent. Otherwise, NAEs prove to be trapped between the shore and the persistent RC jet, so that any contaminants captured by the CNAEs are to be accumulated and transported along the coastline leading to significant contamination of coastal waters and beaches. The CNAEs' negative role has been scrutinized in simulations with a deepwater oil blowout conducted in a prospective region for oil drilling in the Black Sea.

The present work illustrates what may happen in the case, when the Caucasian anticyclonic eddy are formed in the region between Sukhumi and Sochi moves along the Caucasian coast and, being squeezed between the RC and coast, will arrive in the near-shore region south of the Kerch Strait. In this particular case, the CNAE will entrain oil coming from the deepwater source, accumulate and deliver it on the coast along a track of the CNAE.

As satellite observations indicate, sometimes a CNAE can suddenly stop on its way along the Caucasian coast (it often happens in the region between Novorossiysk and Sochi), and, rapidly growing, protrudes into the RC creating a large anticyclonic

system consisting of an anticyclonic meander and CNAE. Such a system was revealed to be unstable and, as satellite observations (*Zatsepin et al., 2003*; *Korotaev et al., 2003*) and numerical simulation (*Korotenko, 2017*) evidenced, protruding deeply into meander. It leads to the rupture of the meander by the eddy and, thus, detachment of the latter from the RC. Such a sequence of events is certain to have a great impact on the behavior of the submarine oil plume and will be investigated in the future works.

## ACKNOWLEDGEMENTS

The author is grateful to D. Dietrich and M.J. Bowman for their help with the hydrodynamic model setup and tuning, as well as to three anonymous referees for their valuable remarks and suggestions that significantly helped to improve the paper.

### Funding

This work was supported by the Russian Scientific Foundation (No. 14-50-00095 and 18-17-00156). The funders had no role in study design, data collection and analysis, decision to publish, or preparation of the manuscript.

### Grant Disclosures

The following grant information was disclosed by the authors:
Russian Scientific Foundation: No. 14-50-00095 and 18-17-00156.

### Competing Interests

The author declare that they have no competing interests.

### Author Contributions

- Konstantin A. Korotenko conceived and designed the experiments, performed the experiments, analyzed the data, contributed reagents/materials/analysis tools, prepared figures and/or tables, authored or reviewed drafts of the paper, approved the final draft.

### Data Availability

Architecture of the oil spill model is presented in Fig. 3 (it is the same as of NOAA GNOME https://response.restoration.noaa.gov/oil-and-chemical-spills/oil-spills/response-tools/gnome.html).

DieCast ocean circulation model code is described at http://efdl.as.ntu.edu.tw/research/diecast/MANUAL/users_manual/.

### Supplemental Information

Supplemental information for this article can be found online at http://dx.doi.org/10.7717/peerj.5448#supplemental-information.

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
