# Peer review of "Effects of mesoscale eddies on behavior of an oil spill resulting from an accidental deepwater blowout in the Black Sea: an assessment of the environmental impacts"

_PeerJ, doi:10.7717/peerj.5448_

## Round 0.1 · original submission · Major Revisions

Dear Konstantin,

Three reviewers have commented on your above paper. Although they saw some merit in your work, they also indicated several criticisms suggesting that it needs major revisions before a decision can be made on its suitability for publication in PeerJ. Please address all the issues raised in the revision and include such comments in the discussion of the manuscript.

Reviewer 1 ·

Basic reporting

Basic reporting and general comments
The manuscript does not show clear, unambiguous, professional English language used throughout.
Intro & background show context. Literature is well referenced & relevant.
However, the manuscript is not well structured.
Some figures contain mistakes (e.g., Fig. 2, 7) and do not have high quality (e.g., Fig. 6, 9, 11). Figure captions are not always exhaustive (e.g., Fig. 6, 8, 9).
Some equations and model specifications are represented in a confusing way (e.g., Eq. 2. and boundary conditions in string 337).
Research question is well defined, relevant & meaningful. It is stated how the research fills an identified knowledge gap. But permanent pycnocline as an important Black Sea feature, that influences the vertical droplet transport, is not mentioned.
Rigorous investigation is performed to a high technical & ethical standard.
However, modeling method is not transparent, which makes it impossible to replicate.

Specific Comments
1. String 111: Please explain the connection between cold methane seeps (yellow squares in Fig. 1) and possible deepwater oil blowouts.
2. String 237: Which diameter droplets do raise at 0.07 ms-1?
3. String 251: Please focus on the processes modeled by you not by Sokolofsky. Point out that near-field modeling the buoyant jet is beyond your model scope.
4. String 275: Where is buoyancy flux?
5. String 306: Please draw the profile/s of V3(x3) and K3(x3).
6. Figure 2: Please correct: Bosphorus, Kizil Irmak. What are dotted arrows on the NW shelf denoted?
7. Figure 7, 10: Please correct scale bar.
8. Figure 9, 11: Please redraw in color.
9. Some figures (e.g., Fig. 1-3, 8) have been already published e.g., in Korotenko (2016). Please insert the reference to the published graphics.

Experimental design

My opinion is expressed in the previous section.

Validity of the findings

My opinion is expressed in the previous section.

Additional comments

My opinion is expressed in the previous section.

Reviewer 2 ·

Basic reporting

No comment

Experimental design

No comment

Validity of the findings

No comment

Additional comments

1. In the introduction the author should refer briefly to the “Oil Platform Leaks”challenge of the EU EMODNET Black Sea Check point activities, which are of relevant to this ms.

2. In deep oil spill model section, provide the corresponding API number of the used light oil type for the simulations. This is important, especially with the Fig. 11.

3. The sentence in lines 233-234 to be modified as: “It is apparent from Eq. 1, the more size of an oil droplet and greater density difference between oil and ambient water the more terminal velocity of the droplet”.

4. Provide an additional plot, similar to the Figure 11, indicating the % of the modeled fate parameters.

5. In line 321 correct the typo error : “released”

6. While in line 335 it is mentioned that the evaporation act within the first few days, in the Fig. 11 the plot concerning the evaporation shows that the evaporation is a process increased by time, more than few days. Explain or correct.

7. In lines 382-384, indicated the period/years of the used climatological T and S data for the hydrodynamical simulations.

8. In lines 470-474, the estimated time (3.7 hours) needed for the first oil drop to reach the sea surface from ~1000 meter seems to be large. I propose to look the recent paper by Lardner and Zodiatis 2017 mentioned below.

9. Studies regarding the oil plumes modeling and oil spill modeling in the Black Sea can be found in the following papers and I suggest to the author to have a look and include them at the references:

Lardner R. and G. Zodiatis (2017). Modelling oil plumes from subsurface spills, Marine Pollution Bulletin , http://dx.doi.org/10.1016/j.marpolbul.2017.07.018

Zodiatis G. and Lardner, R. and Solovyov, D. and Panayidou, X. and De Dominicis, M. (2012). Predictions for oil slicks detected from satellite images using MyOcean forecasting data. Ocean Science, Volume 8, Pages 1105-1115.

Yapa, Poojitha, Zheng, Li, 1997. Simulation of oil spills from underwater accidents I:
model development. J. Hydraul. Res. 35, 673–687.

Zheng, Li, Yapa, Poojitha, 1998. Simulation of oil spills from underwater accidents II: model verification. J. Hydraul. Res. 36, 117–134.

Reviewer 3 ·

Basic reporting

Language and Length of paper

There are some language problems. Already in the title:
Effects of mesoscale eddies on behavior and environmental impacts of oil spilled in result of an accidental deepwater blowout in the Black Sea
The phrase “In result” – should usually be “as a result”. E.g. [Title, plus Line 42, 65, 116, 141…]
I suggest: Effects of mesoscale eddies on THE behavior of oil spilled AS A RESULT of an accidental deepwater blowout in the Black Sea and its environmental impacts

[Lines 12-13]: employing numerical modeling is not exactly ‘easy’. I would suggest to cut that word, and rewrite the sentence.
Suggest: Employing numerical modeling as a prediction tool is one of the most efficient methods to understand oil spill behavior under various environmental forces.

[Lines 27-28] What does a 20-day scenario tell us? Explain that this is the time needed to see CNAEs evolve, and oil gets captured and transported by the AEs.
[Line 76]: “disastrous events, they seem to be inevitable due to a number of reasons”. …Such as? What could be a typical reason for a disastrous event in the Black Sea? This is not the same as e.g. the Gulf of Mexico, where many hurricanes hit.
[Line 79-82]: “The Black Sea is a water body that, owing to its semi-isolation from the open ocean, suffers from strong ecological disequlibria caused by pollution arising from many contaminants”,. Explain clearer what the disequlibria lie in? Rare flushing, strong stratification, …, thus contaminants may have a long-term effect…
[Line 83-88]: this should be rewritten, it does not flow well. Also, you mention that there are very many natural seeps in the BS, so obviously not only accidental oil spills posed a problem in the past.

[Lines 122-123]: repetition from the abstract.
[Lines 126-129]: repetition from the abstract

[Line 361]: high-resolution

I think that the scenarios chosen could be mentioned earlier, the introduction is too long and wordy. Tighten the introduction in order to get to the main interest of the paper. In general the paper is too long considering it’s in reality only describing a couple of scenarios

Experimental design

I cannot see that this paper lies within the scope of the PeerJ journal. Oil spill modeling is a physical/chemical discipline that involves numerical geophysical modeling of atmosphere, ocean, and waves, as well as a knowledge of oil chemistry and oil weathering processes.

The paper aims to address impacts on the marine environment, but only does so to the extent of evaluating whether certain regions of the coastlines of the Black Sea might be struck by oil in case of an accidental blowout under certain weather conditions. Even if this is an interesting subject matter, the study cannot really say anything about environmental impacts as long as it does not link to any scientific knowledge about the particularities of specific vulnerable ecosystems or fragile water systems around the Black Sea.

As such, I am not sure that this paper will find an interested audience within this particular journal. The author has previously published in scientific journals such as “Oceanology”, “Spill Science and Technology Bulletin”, “Terrestrial, Atmospheric and Oceanic Sciences”, “Journal of Marine Systems”, etc. Although Peer J aims to cover Environmental Sciences, the topics therein do not cover geophysics, but rather biology and ecology.

I would say that the methods have been sufficiently described to replicate, but are lacking in that other models and methods could have been used that would have provided more conclusive results. E.g. statistical runs (meaning many runs over several seasons) and more locations spread along interesting/potential drilling sites. This will hopefully be addressed in future studies.

Since the DieCast model does not include data assimilation, it can be very tricky to forecast the exact placement of eddies in time and space, and as such, the model is better used for process studies. For oil spill contingency and preparedness, it would be useful to have a more rigid assessment of the potential extent of shoreline contamination, combined with maps of ecologically vulnerable parts of the shoreline, as well as practical advice on where to place depots for oil spill cleanup efforts.

The intro and background show context, and the literature referenced is comprehensive, and shows that the author knows the context pretty well with regards to the geophysical modeling. However, the use of the oil spill modeling shows a simplistic approach and a basic knowledge of the field of oil weathering processes that have partly been taken from outdated literature (e.g. the oil is presented as a mixture of 8 hydrocarbon groups following Mackay & McAuliffe, 1988, where removal of oil through the evaporation process is linear over a time of 20 days (see figure 11), whereas in reality, a light, crude oil such as used in this study tends to evaporate away by 25% within the first few hours, 75% within the first few days. (E.g. D. Mackay, R.S. Matsugu, Evaporation rates of liquid hydrocarbon spills on land and water, Can. J. Chem.Eng. 51 (1973) 434-439.)
The mass balance of the oil therefore will necessarily be unrealistic after 20 days, and thus the assessment of the scale of contamination of marine environment, that was one of the aims of the study, will be wrong. Other important weathering processes, in particular emulsification, is not included in the model. As for the natural dispersion, he amount of dispersed oil seems very large. Is your droplet size distribution correct?

[Lines 133-138]: You say that “in deep waters, the movement and fate of a multiphase plume is governed by the gas-oil separation process, rising velocity as well as background currents and stratification, while at the subsurface layer, the plume evolution and its fate experiences the influence of currents induced by local winds, Stokes drift and physicochemical processes which change the oil properties.”

I can’t see how the Stokes drift, or any wave parameters have been used at all in this study, although it is mentioned that waves are input to the to the Deepsea Oil Spill Model (lines 195-195).
I can’t see that you have discussed the effects of vertical mixing on the transportation of the oil, although this is very important for the overall transportation of oil as well as the oil mass budget.

Validity of the findings

The aims of the study are:
1) To “elucidate the behavior of the subsea plume” – a simple plume model is set up, but the scope of the study does not add anything new to the general knowledge of deep sea plume modeling. (Multi phase aspects are lightly skipped, deep sea formation of gas hyrates is not mentioned as a problem in modeling)
2) To assess scales of contamination of marine environment and coastlines in possible accidental blowouts – with two scenarios studied of a hypothetical deep sea oil well blowout placed at one location on the Shatsky Ridge, it is hard to say that any scale of contamination has been assessed. I’d rather say that a model has been set up that might work to do such an assessment in a much wider study.

I would not say that the investigation has been rigorous enough to address the research aims stated in the introduction, but would view it more as an initial effort giving some answers to how well the oil spill model performs.

The role of the CNAEs is illustrated well, in the way a “highly “oiled” eddy would extend and move near the coasts, depositing massive amounts of oil and polluting the Caucasian and Crimean beaches.
Why has only winter situation been studied in the 2 scenarios? [Lines 618-619]. For impact studies, it might be useful to only use climatological winds, and rather study different periods of the year with known differences in circulation patterns

Most of the figures are relevant and clear, but I find some figures missing for comparison between the climatological wind situation and the NCEP daily averaged wind situation. E.g. figure 9 showing successive vertical plots of the plume and surface spread of oil against the wind sticks for NCEP winds. Why not have the same plot for the climatological winds? What does that mean? Are the climatological winds not of high enough resolution in time?
Figure 11 is an important figure, showing the oil mass budget over the 20 day scenario. It would be easier to interpret this figure if it were in colour.

---

## Round 0.2 · Minor Revisions

Dear Author
Before accepting the manuscript for the publication, I kindly ask you to re-submit it after a general revision of the English language is done. You also should answer the points highlighted by reviewer 1.

In particular, I ask you to address the following:
"“5. String 306: Please draw the profile/s of V3(x3) and K3(x3).
A. Let I omit these profiles since for the used version of the model they have not much information but substantially boost the text.
Explanation: (1) Because the vertical velocity throughout the water column of is very small (~10-5 m/s) compared to the droplet terminal velocity, the former will have a little effect on the vertical displacements of oil droplets and everything would be determined by terminal velocity”

and

"(3) V3(x3) and K3(x3) profiles, which are extremely uncertain in the Black Sea and are relevant to the vertical transport of small droplets, are not shown."

Thank you very much

Reviewer 1 ·

Basic reporting

I could not see any major revisions made by the author:
(1) English language is really awful.
(2) There are not any new findings in the manuscript. Horizontal transport of the oil droplets is well known. Any horizontal intrusions of droplets are not visible (Fig.6). Influence of the Black Sea permanent pycnocline is neglected because the rise velocities of droplets are very high, as the author confirms in his response below.

“5. String 306: Please draw the profile/s of V3(x3) and K3(x3).
A. Let I omit these profiles since for the used version of the model they have not much information but substantially boost the text.
Explanation: (1) Because the vertical velocity throughout the water column of is very small (~10-5 m/s) compared to the droplet terminal velocity, the former will have a little effect on the vertical displacements of oil droplets and everything would be determined by terminal velocity”


(3) V3(x3) and K3(x3) profiles, which are extremely uncertain in the Black Sea and are relevant to the vertical transport of small droplets, are not shown.
Unfortunately, my conclusion is to reject.

Experimental design

Everything I want to say is written in the Basic reporting Section

Validity of the findings

Everything I want to say is written in the Basic reporting Section

Additional comments

Everything I want to say is written in the Basic reporting Section

Reviewer 3 ·

Basic reporting

Figures have been improved, and the text is clearer. I assume that PeerJ has an editor that can help improve the English once the paper has been approved for publication.

Experimental design

No comment

Validity of the findings

No comment

Additional comments

I thank the author for his replies to my comments and questions. Several points have been improved and clarified in this latest version of the paper, and choices made have been justified.

---

## Round 0.3 · accepted · Accept

Dear Konstantin

The revised manuscript satisfies the editor's requests. Thus I suggest it be published.

Many thanks